# Sea ice and productivity changes over the last glacial cycle in the Adélie Land region, East Antarctica, based on diatom assemblage variability

Lea Pesjak[1], Andrew McMinn[1], Zanna Chase[1], Helen Bostock[2,3]

[1]Institute for Marine and Antarctic Studies, University of Tasmania, Hobart, 7000, Australia
[2]University of Queensland, Brisbane, 4072, Australia
[3]National Institute of Water and Atmospheric Research (NIWA), Wellington, New Zealand

*Correspondence to*: Lea Pesjak (lea.pesjak@utas.edu.au)

**Abstract**

Although diatoms can provide important paleoenvironmental information about seasonal sea ice extent, productivity, sea surface temperature and ocean circulation variability, there are still relatively few studies analysing the last glacial cycle near the Antarctic continent. This study examines diatom assemblages over the last glacial cycle from core TAN1302-44, offshore Adélie Land, East Antarctica. Two distinct diatom assemblages were identified using principal components analyses. The PC 1 assemblage is characterized by *Thalassiosira lentiginosa, Actinocyclus actinochilus, Eucampia antarctica*, *Azpeitia tabularis* and *Asteromphalus hyalinus*, and is associated with the interglacial, sedimentary Facies 1, suggesting that the MIS 5e and Holocene interglacials were characterised by seasonal sea ice environments with similar ocean temperature and circulation. The PC 2 assemblage is characterized by *Fragilariopsis obliquecostata, Asteromphalus parvulus,* and *Thalassiosira tumida,* and is associated with the glacial, Facies 2. The variability of PC 2 indicates that during the MIS 4-2 glacial and the last glaciation there was an increase in the length of the sea ice season compared with the interglacial period, yet still no permanent sea ice cover. The initial increase of PC 2 at the start of the glaciation stage and then a gradual increase throughout late MIS 4-2, suggests that sea ice cover steadily increased reaching a maximum towards the end of MIS 2. The increase in sea ice during glaciation and MIS 4-2 glacial is further supported by the increase in the *Eucampia* index (Terminal/ Intercalary valve ratio), an additional proxy for sea ice, which coincides with increases in PC 2. Aside from the statistical results, the increase in the relative abundance of *Thalassiothrix antarctica* at 40 cm and 270 cm suggests that during the last two deglacials there was a period of enhanced nutrient delivery, which is inferred to reflect an increase in upwelling of Circumpolar Deep Water. Interestingly, the diatom data suggest that during the last deglacial, the onset of increased Circumpolar Deep Water occurred after the loss of a prolonged sea ice season (decrease in PC 2), but before the ice sheet started to retreat (increase in IRD). Together, these results suggest the changes in sea ice season potentially influenced the ocean's thermohaline circulation and were important factors in driving the climate transitions. The results contribute to our understanding of the sea ice extent and ocean circulation changes proximal to East Antarctica over the last glacial cycle.

## 1 Introduction

Ocean circulation near Antarctica's ice sheets is changing under the influence of climate change (Pritchard et al. 2009; Depoorter et al. 2013; Alley et al. 2015; Silvano et al. 2018; Rignot et al. 2019; Minowa et al. 2021). The two significant parameters in the atmosphere-ocean-ice sheet interaction system are duration and extent of seasonal sea ice and the ocean's thermohaline circulation. Antarctic sea ice is recognised as an important driver of climate, as it affects the $CO_2$ exchange between the Southern Ocean and the atmosphere (Crosta et al. 2004;

Kohfeld & Chase 2017), planetary albedo, and the ocean's thermal gradients (Gersonde & Zielinski 2000). Locally, its seasonal variation can affect ice shelves, increasing melting at the marine edge (Massom et al. 2018), ultimately destabilising the ice sheet (Pritchard et al. 2012). Furthermore, its seasonal expansion and retreat influences primary productivity; by limiting light, thus decreasing productivity, although meltwater can also stimulate phytoplankton blooms (Knox 2006). The second significant parameter affecting climate is the

ocean's thermohaline circulation. On the Antarctic margin this is driven by the formation of Antarctic Bottom Water (AABW), and the upwelling of Circumpolar Deep Water (CDW). Modern observations suggest that Antarctic ice sheet melt rates increase with enhanced upwelling of CDW (Pritchard et al. 2012; Rignot et al. 2019; Minowa et al. 2021) and this causes a decrease in the production of AABW (Williams et al. 2016; Silvano et al. 2018), which may further influence ice sheet melt (Silvano et al. 2018). Understanding the past changes in

sea ice and oceanography proximal to Antarctica, especially during past climate transitions and warmer than present interglacials, such as the last interglacial, MIS 5e, may provide further insight into the mechanisms of atmosphere-ocean-ice sheet interaction, to predict future changes and provide analogues for future outcomes under a warming climate (Masson-Delmotte et al. 2013).

Studies of diatom assemblages from ocean sediments can be used to reconstruct past ocean environments, including the extent and duration of seasonal sea ice, surface ocean circulation, and productivity (Cooke & Hays 1982; Pichon et al. 1992; Taylor & McMinn 2001; Crosta et al. 2004; Gersonde et al. 2005; Armand et al. 2005). Diatom studies are based on the identification and quantification of individual species and groups of species, which are used to reconstruct paleoenvironments based on an understanding of the species' modern

habitat (Table S1) from both water column (Medlin and Priddle 1990; Ligowski 1992; Moisan and Fryxell 1993) and from surface sediment studies (Zielinski and Gersonde 1997; Armand et al. 2005; Crosta et al. 2005). However, the interpretation can be influenced by processes such as selective dissolution within the water column and/ or sediment (Warnock and Scherer 2015; Shemesh, Burckle & Froelich 1989; Zielinski & Gersonde 1997), winnowing of lighter species' valves by bottom currents (Taylor, McMinn & Franklin 1997;

Post et al. 2014), or variable influx of terrigenous matter (Kellogg & Truesdale 1979; Schrader et al. 1993). Therefore, when reconstructing the past environment, it is important to consider all these processes.

There are many diatom-based studies of the interglacials, especially from the Holocene period from the Antarctic continental shelf (McMinn 2000; Taylor and McMinn 2001; Leventer et al. 2006; Crosta et al. 2007;

Maddison, Pike & Dunbar 2012; Peck et al. 2015; Mezgec et al. 2017; Torricella et al. 2021). However,

advanced ice sheets, or permanent sea ice, in past glacials led to no diatom productivity over the Antarctic continental shelf, and reduced productivity over the slope (Pudsey 1992; Lucchi 2002; Hartman et al. 2021). Additionally, advancing ice sheets would have removed most of the glacial sediment record from the continental shelves (Domack 1982; Escutia et al. 2003). This may be one of the reasons why there are so few studies from

proximal Antarctica detailing the composition of diatom communities during glacial periods and over the last glacial cycle (Caburlotto et al. 2010; Holder et al. 2020; Hartman et al. 2021; Li et al. 2021; Chadwick et al. 2022).

Overall, limited previous paleoenvironmental studies based on diatoms from the Antarctic continental slope

suggest that, during the last glacial cycle, there was seasonal sea ice cover over the Adélie region (Caburlotto et al. 2010), a permanent sea ice cover in the Western Ross Sea (Tolotti et al. 2013), and a prolonged sea ice season in several regions including offshore Cape Adare, the Ross Sea (Hartman et al. 2021), offshore Enderby Land (Li et al. 2021) and offshore the Sabrina Coast (Holder et al. 2021). However, persistent biological productivity has been recorded from offshore Cape Adare (based on diatom studies; Hartman et al. 2021), and

the Weddell Sea (based on studies of foraminifera; Smith et al. 2010). These blooms have been suggested to represent localised polynyas (Arrigo & Van Dijken 2003) that existed during the last glacial. Only a couple of studies have looked into climate transitions during the last glacial cycle on the Antarctic margin. They show that during the last deglacial there was a decrease of the sea ice season and an increase in upwelling of CDW over the Enderby Land and Ross Sea continental margin (Li et al. 2021; Tolotti et al. 2013), while the last glaciation

stage is reported to comprise oscillations in the sea ice season offshore Cape Adare (Hartman et al. 2021). Here we use diatom assemblages to understand the changes in the duration of the sea ice season and in CDW upwelling in the Adélie region over the last glacial cycle, including the glaciation and deglacial transitions.

## 2 Materials and Methods

### 2.1 Site description

Core TAN1302-44 (Tan_44) was recovered using a gravity corer with a 2-tonne head, from the WEGA channel,

on the continental slope north of Adélie Land and the George Vth Land coastline (Adélie region), at 64°54.75 S, 144°32.66 E, from 3,095 m depth (Fig. 1) by R/V Tangaroa in February 2013 during voyage TAN1302 (Williams 2013). The location is ~100 km north off the continental shelf break. The core site is located within the modern seasonal sea ice zone (Fetterer et al. 2017), covered by sea ice from April to November each year (Fig. 1, Spreen, Kaleschke & Heygster 2008). The major oceanographic features of this region, which directly

influence the site (Caburlotto et al. 2006; Williams et al. 2008), include Adélie Bottom Water (Adélie AABW), which forms below ~2,000 m from mixing of cooler Dense Shelf Water (DSW) formed on the shelf with warmer and nutrient rich CDW, and the wind-driven, westward flowing Antarctic Slope Front (ASF; Jacobs 1991; Williams et al. 2008; Fig. 1). The Antarctic Circumpolar Current (ACC), depicted in Fig 1. (Southern

Boundary of the Antarctic Circumpolar Current front), does not influence the core site, but the ACC has a
significant influence over Southern Ocean productivity and diatom species distribution (Supplement, Table S1).

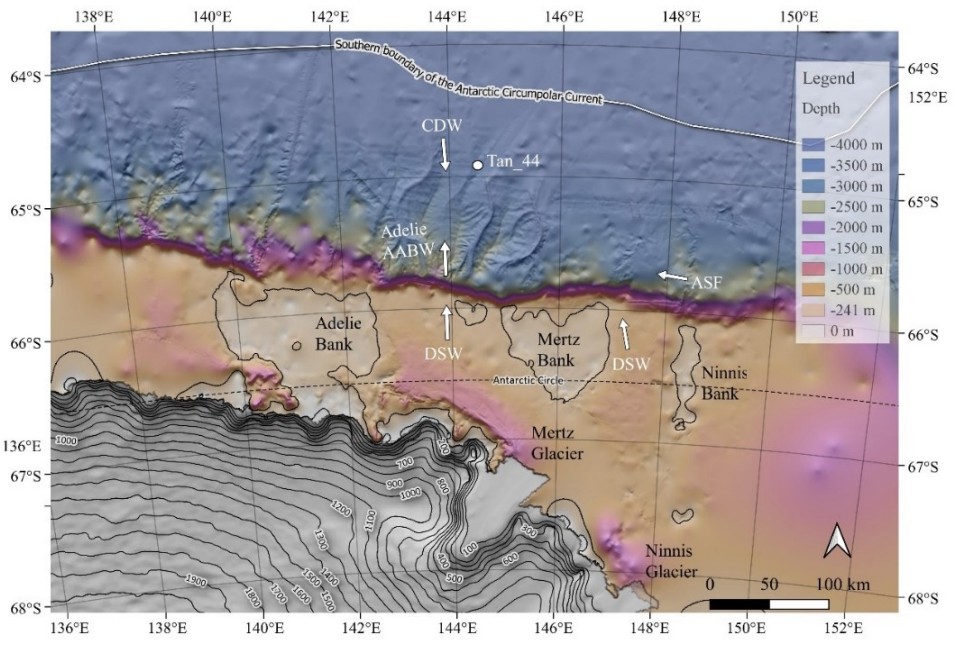

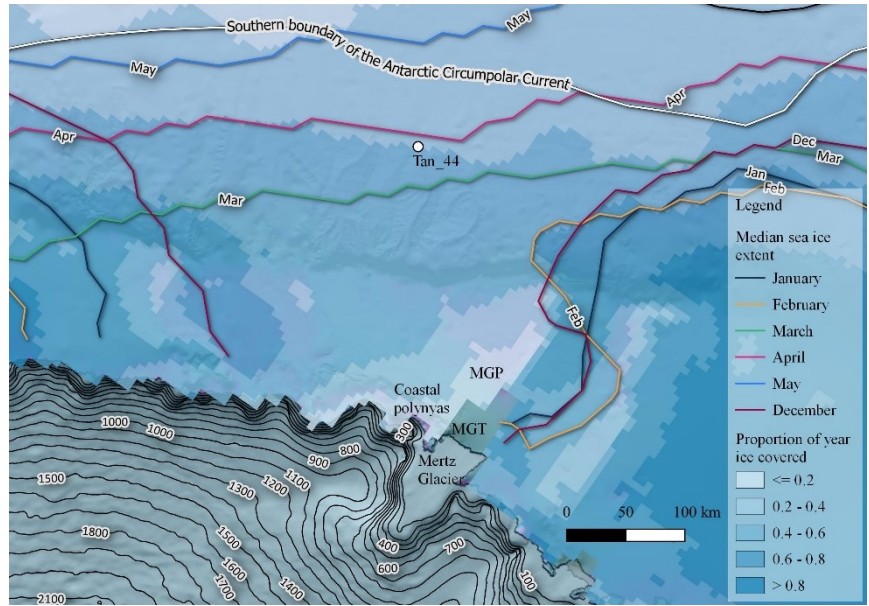

**Figure 1** *Top*: location of core Tan_44 with respect to the regional bathymetry (Arndt et al. 2013);
oceanography (Orsi et al. 1995; Williams et al. 2010) and cryosphere (Helm et al. 2014). The Mertz and Ninnis
glaciers (Helm et al. 2014) are dominant glacial features in the region. Adélie and Mertz banks are prominent
geomorphological features on the continental shelf, while deep channels are prominent on the continental slope.
Tan_44 is located within the WEGA channel. The site is influenced at present by Adélie sourced Antarctic
Bottom Water (Adelie AABW), circumpolar deep water (CDW), and along slope flow of Antarctic Slope Front
(ASF; Williams et al. 2010). *Bottom*: modern seasonal sea ice cover: the darkest blue showing regions covered
by ice for more than 80% of the year, while the lightest blues indicates areas covered by ice less than 20% of the
year (Spreen, Kaleschke & Heygster 2008), indicating the Mertz Glacier Polynya (MGP) and the coastal
polynyas, where the main proportion of Adélie DSW forms, west of the Mertz Glacier Tongue (MGT; Williams
et al. 2010). The figure also shows the core site is covered by sea ice from April to November each year
(coloured lines; Fetterer et al. 2017).

## 2.2 Biogenic silica


Biogenic silica is used in this study as an indicator of paleoproductivity (Bonn et al. 1998; Wilson et al. 2018). Analyses of biogenic silica were undertaken at 20 cm intervals down Tan_44. This study uses a modified wet-leaching technique (Mortlock and Froelich 1989; and DeMaster 1981), based on the premise that dissolution of fragile diatom tests is more rapid than the dissolution of silica from non-biogenic sources e.g. quartz grains. The

time-series approach introduced by DeMaster (1981) was used. For quality control, two in-house standards were used from the Chilean and the Antarctic margin (Tooze et al. 2020). If the silica concentrations of the standards or the samples decreased with time during the hourly measurements, the whole experiment was repeated. The overall reproducibility of the method, assessed as the relative standard deviation of the standards, was +/-7%.

## 2.3 Si/Al (XRF)


Si/Al is used in this study as an indication of biogenic silica, and therefore paleoproductivity (Rothwell and Croudace 2015). X-ray fluorescence scanning (XRF) was completed at 2 mm resolution using an ITRAX scanner (Gadd and Heijnis 2014) at the Australian Nuclear Science and Technology Organisation (ANSTO).

The scanning was performed on u-channel sub-samples of the cores (of dimensions 2x2 cm, 1 m-long sections), which were stored in plastic containers and covered by thin plastic film. Anomalous spikes in data, identified by eye as significant increases or decreases occurring on mm-scale, were removed. The data was then smoothed using a 3-point running average.

## 2.4 Ice Rafted Debris (IRD)


Increased ice rafted debris (IRD) are used in this study as indicators of past Antarctic ice sheet retreat, and interglacial periods (Grobe et al. 1992; Cook et al. 2013; Patterson et al. 2014). IRD analysis was completed using two methods, counting visible grains from X-radiographs (grains ≥1 mm, in 5 cm sections), and counting

sieved grains (> 500 μm) per dry weight of total sample. The size >500 μm, medium sand (Patterson et al. 2014) was chosen as the size that defines IRD because laser particle diffraction of samples showed the grain size <250 μm forms the matrix of all the samples. This is in contrast to other Antarctic studies, which have defined IRD using a range of different sizes from >2 mm (Grobe et al. 1992; Diekmann et al. 2003), very coarse sand size, >1 mm (Lucchi et al. 2002; Pudsey & Camerlenghi 1998), >250 μm (Wilson et al. 2018), and >125 μm

(Cook et al. 2013; Passchier 2011).

## 2.5 Facies and age model

Radiocarbon dating was undertaken to support age model development, using the Acid Insoluble Organic Matter

(AIOM) method, conducted at ANSTO, Sydney in April 2017 according to Hua et al. (2001); Fink et al. (2004); and Stuvier and Polach (1997). The raw radiocarbon ages were calibrated using CALIB, version 7.1, Marine13

calibration curve (Reimer et al. 2013), and the regional variation to the global marine reservoir correction, ΔR, of 830 yrs ±200 yrs, following previous work done in this region by Domack et al. (1989).

A facies model was developed using the lithological unit characteristics (Supplement S1.2.1) and the combination of other data, primarily biogenic silica, Si/Al, and ice rafted debris (IRD). The definition of facies was designed to capture large variability in physical and geochemical quality of sediment, including large changes in productivity (biogenic silica, Si/Al and Ba/Ti) and sedimentology (IRD content; Wilson et al. 2018; Salabarnada et al. 2018; Wu et al. 2017; Bonn et al. 1998; Grobe & Mackensen 1992; Patterson et al. 2014).


The age model of Tan_44 is based on the facies model and two radiocarbon dates from the top 25 cm of the core, using the premises that variability in facies, including large changes in productivity proxies (biogenic silica, Si/Al and Ba/Ti) and IRD content, present glacial to interglacial climate variability (Wilson et al. 2018; Salabarnada et al. 2018; Wu et al. 2017; Bonn et al. 1998; Grobe & Mackensen 1992; Patterson et al. 2014).


**2.6 Diatom counts and Shannon Wiener biodiversity index**

Diatom species were counted from samples taken every 10 cm down core Tan_44 (starting at 5 cm, then 20 cm). The samples were processed following the methods outlined in Taylor & McMinn (2001). A small section of the

sediment core (<0.5 cm thick) was soaked in 15% hydrogen peroxide overnight, to remove organic matter and to disaggregate any clay. The samples were rinsed with deionised water through a 100 μm and a 10 μm sieve, in order to obtain a >10 μm, <100 μm grain fraction. This fraction was left overnight to settle. Excess water above the sample was pipetted out, and the remaining sample was stored in a 100 ml tube. A drop from each shaken tube was pipetted onto a glass cover slip over a hotplate at 50°C, to evaporate excess water. The samples were

then mounted with Norland Optical Adhesive 61 and cured in sunlight. Diatom identification and counts were undertaken using a Nikon light microscope (Eclipse Ci, DS-Ri2) at 1000 X magnification. Each sample was traversed until >400 valves were counted. Broken valves that were >50% complete were included in the count and in the case of elongated species, such as *Thalassiothrix* and *Trichotoxon* that are subject to fragmentation, only the ends were counted (McMinn et al. 2001). Lower valve numbers, of less than 400 valves per slide, were

encountered in samples at 80 cm, and from 350-300 cm. The numbers of valves within the 350-320 cm samples were extremely low (7-16 valves per slide). Due to the scarcity of valves in these samples, and well below 400 valves per slide observed within 320-300 cm, only samples from 290-5 cm are included in statistical analysis.

Some species were grouped together due to morphology and habitat indicators, these groups are the

*Fragilariopsis* group, comprising *F. obliquecostata*, *F. sublinearis*, *F linearis* and *F. cylindrus*, the *Thalassiothrix* group, comprising *Thalassiothrix antarctica, Thalassiothrix longissima* and *Trichotoxon reinboldii, and the Rhizosolenia* group*, comprising Rhizosolenia antennata* var. *semispina , Rhizosolenia antennata* var. *antennata, R. simplex, R. polydactyla var polydactyla, Rhizosolenia sp.,* and *Proboscia inermis.*

The relative abundance of each species (or group) was expressed as the number of valves of that species divided
by the total valve count (expressed as %). Species or species groups with >1.8% in at least two samples were
included in statistical analysis, except in case of the *Fragilariopsis* group, which apart from *Fragilariopsis
obliquecostata* (present at >1.8% in at least 2 samples) also included much rarer sea ice species (*F. sublinearis,
F. linearis, and F. cylindrus).*

Species or species groups present at >1.8% in at least 1 sample, and thus excluded from statistics, were included
in results and discussion due to their environmental indications (Table S1). These species include the
*Thalassiothrix antarctica* group, represented mainly by *Thalassiothrix antarctica* , and the *Rhizosolenia* species
group, presented mainly by *Rhizosolenia antennata* var. *semispina* and *Rhizosolenia antennata* var. *antennata*
(the sum of both species was up to 1.2-1.6% in three samples). The *Thalassiothrix* group is also discussed where
there is a significant increase in broken valves, yet the relative abundance (i.e., counted valve ends) is 0%.

The *Eucampia* index was used as an indicator of sea ice presence (Fryxell et al. 1991). It represents the ratio of
the number of terminal valves to the number of intercalary valves of *Eucampia antarctica* species, and its
increase is associated with more sea ice in the environment. In the open ocean the *Eucampia antarctica* species
grow in longer chains, while in sea ice waters they grow in shorter chains (Fryxell 1991). The chains comprise
intercalary valves in the middle, and terminal valves at the ends, and therefore, the more terminal valves, the
more sea ice (Fryxell 1991: Kaczmarska et al. 1993). The *Eucampia* index was only calculated where the total
*Eucampia antarctica* count was 100 valves and above.

An assessment of the diversity of diatoms in each sample was determined using the Shannon-Wiener diversity
index. The Shannon-Wiener diversity index (Spellerberg et al. 2003) was calculated according to the formula:

$$H = - \sum_{i=1}^{n} [p_i \; x \ln p_i]$$

**2.7 Statistical analyses: cluster analysis and principal component analysis**


The relative diatom abundance data set was analysed using a hierarchical cluster analysis and principal
component analysis (PCA), in Statistical Package for Social Sciences (SPSS) software package. For these
analyses the relative abundance data was logarithmically transformed using the equation: Abundance = $\log_{10}$
(x+1), where x= relative abundance (%), (Taylor, McMinn & Franklin 1997). Cluster analysis (Burckle 1984;
Truesdale & Kellogg 1979) involved calculating the average distance between groups. The PCA (Taylor,
McMinn & Franklin 1997; Zielinski & Gersonde 1997) was undertaken in two stages. In Q-mode, investigating
relationships between variables (species), and in R-mode, analysing relationships between samples (Shi 1993).
The factor variance used to extract the number of components for Q-mode analysis was established at ≥12%

variance. The factor variance used to extract the number of components for R-mode analysis was established at
≥42%. Factor variance is the amount of the total variance of all of the variables accounted for by each
component (factor) (https://www.ibm.com/docs/en/spss-statistics). Outputs from both Q and R analyses
underwent a Varimax rotation. Rotation maintains the cumulative percentage of variation explained by the
chosen components, but the variation is spread more evenly over the components
(https://www.ibm.com/docs/en/spss-statistics). Finally, to demonstrate the strength of the correlation between
the components and productivity proxies (Si/Al and biogenic silica), bivariate Pearson Correlation analyses
were undertaken using SPSS.

## 3 Results

### 3.1 Biogenic silica, Si/Al, ice rafted debris, and the facies model

The facies model is comprised of four facies which alternate down core (Supplement Fig. S1; Table 1). The
main parameters determining the facies (biogenic silica, Si/Al and IRD) are described below. The interpretation
of the facies is further described in the Age Model section.

Biogenic silica varied from 0-22% (Fig. S1; Table 1; Fig. 2; Fig. 3). The highest values were found in the top 40
cm (10-22%), at 260-140 cm (12-16%), and at the base of the core at 540-520 cm (3-11%), coinciding with
olive/grey sandy mud (Facies 1) and olive grey mud (Facies 1A). Moderate to low values (3-10%) occurred in
olive mud (Facies 2A) and grey mud (Facies 2), with the exception of 18% at 140 cm within Facies 2 (Fig. S1;
Fig. 3).

Si/Al (XRF) values varied from 14-28 (Fig. S1; Fig. S2; Table 1; Fig. 2; Fig. 3). Higher values occurred within
olive/ grey sandy mud (Facies 1), while lower values occurred within olive mud (Facies 2A), olive grey mud
(Facies 1A), and grey mud (Facies 2).

High counts of ice rafted debris (IRD; 4-36 grains/5 cm) are found in olive/ grey sandy mud (Facies 1; Fig. S1;
Table 1; Fig. 2; Fig. 3), with maximum counts found at 15-10 cm, at 255-250 cm and at 500-495 cm. Lower
numbers of IRD (0-14 grains/5 cm) are found in grey mud (Facies 2), olive grey mud (Facies 1A) and olive mud
(Facies 2A).

**Table 1 Summary of the characteristics of the four facies present in core Tan_44.**

| CHARACTERISTICS: | FACIES: | | | |
|---|---|---|---|---|
| | 1) OLIVE SANDY MUD | 2A) OLIVE MUD | 2) GREY MUD | 1A) OLIVE GREY MUD |
| Colour | Olive; grey (base layer) | Olive | Grey | Olive grey |
| Structure | Massive; bioturbation; rare laminae | Massive; bioturbation; | Massive; bioturbation; laminae; traction structures | Massive; bioturbation; |
| IRD (grains/5 cm) | 0-36 | 0-10 | 0-14 | 1-15 |
| IRD (grains/g) | 2-15 | 0-1 | 0-1 | 0-2 |
| % Vf-f sand | 1-19 | 1-7 | 0-5 | 0-6 |
| % Vc silt | 8-27 | 3-18 | 4-13 | 7-17 |
| Zr/Rb | 0.6-2.3 | 0.5-1.9 | 0.4-1.4 | 0.7-1.4 |
| % Biogenic silica | 3-22 | 4-10 | 3-18 | 10-11 |
| Si/Al | 15-28 | 14-23 | 12-20 | 16-21 |
| Ba/Ti | 0.01-0.06 | 0-0.06 | 0-0.04 | 0.03-0.05 |
| INTERPRETATION: | Interglacial | Deglacial | Glacial | Glaciation |

## 3.2 Radiocarbon dates and the age model

The two top radiocarbon dates (Fig. 2) indicate the top of the core was deposited between 16.2- 5.2 ka, suggesting Facies 1 is of Holocene and Facies 2A of Late Pleistocene age. Facies 1 at 270-230cm is interpreted as being the last interglacial, MIS5e, supported by the fact that we did not observe any *Rouxia leventerae* (last occurrence at MIS6-5e boundary; (Zielinski and Gersonde 2002) at these depths. The two deeper radiocarbon dates (Table 2) were not included in the interpretation because they imply an unreasonably high sedimentation rate. Similar unreasonable C-14 dates at similar core depths, and even age reversals, were observed in other cores from the region (Pesjak 2022).

The main characteristics of the facies found in Tan_44 (Table 1) in combination with the radiocarbon dates at the top of core, suggest glacial to interglacial variability influenced the productivity proxies (biogenic silica, Si/Al and Ba/Ti; Wilson et al. 2018; Salabarnada et al. 2018; Wu et al. 2017; Bonn et al. 1998; Grobe & Mackensen 1992) and IRD (Patterson et al. 2014; Fig. S1; Fig. 2), which show a strong coincidence with the global benthic $\delta^{18}O$ stack (Lisiecki & Raymo 2005).

**Table 2** Radiocarbon dating (AIOM) conventional and calibrated results.

| Lab No. | Sample | Depth (cm) | Conventional radiocarbon age (yr BP) | Error ± (yr) | δ 13C | Calibrated age (cal. yr BP); ΔR=830+/-200 | Calibrated mean (yr BP) |
|---|---|---|---|---|---|---|---|
| OZV390 | Tan44_0cm | 0.5-3.5 | 5,765 | 45 | -25 | 4,971-5,478 | 5,233 |
| OZV391 | Tan44_25cm | 25.5-26.5 | 14,660 | 80 | -23.6 | 15,837-16,468 | 16,160 |
| OZV392 | Tan44_35cm | 35.5-36.5 | 18,470 | 90 | -23.9 | 20,504-21,082 | 20,803 |
| OZV393 | Tan44_45cm | 45.0-46.0 | 19,150 | 140 | -25 | 21,340 - 22,007 | 21,682 |

.

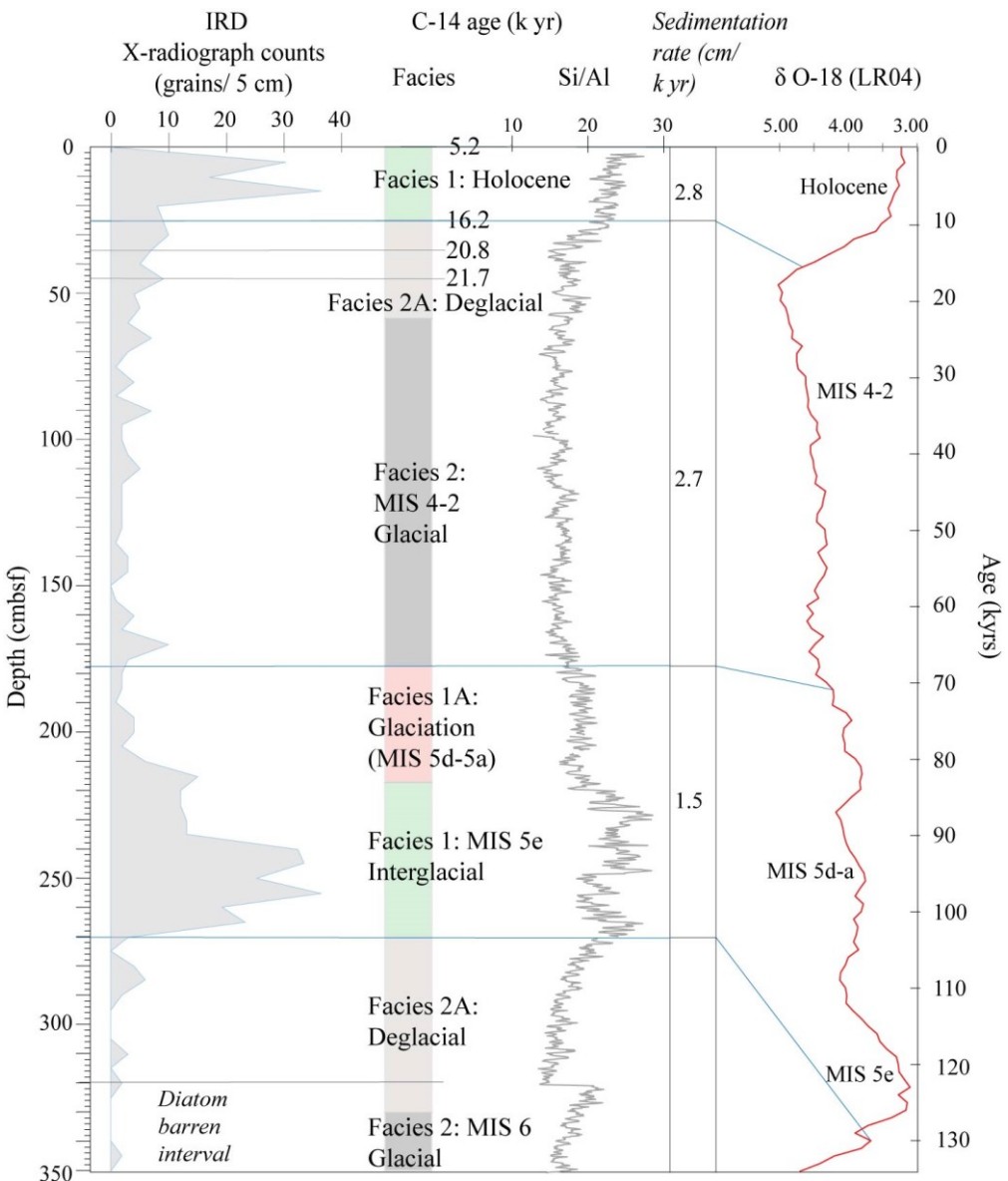

**Figure 2** IRD counts, facies model and Si/Al for core Tan_44. The age model is based on the facies model and supported by radiocarbon dating of two depths at the top of the core, and the comparison of Si/Al to the global benthic stable isotope stack LR04 (Lisiecki and Raymo 2005).

### 3.3 Species distribution, biodiversity, and abundance

All samples contained well-preserved diatom assemblages with little evidence of dissolution, such as frustule thinning. Of the 52 species identified in 34 samples (Table S4), 24 species (Table S1) were included in the species distribution description and of those, 15 species were included in statistical analysis (12 species with an additional 3 included as part of the *Fragilariopsis* group; Fig. 3). One extinct species *Actinocyclus ingens* (Cody et al. 2008) was found at 11% at 60 cm, and at 3% at 280 cm, both at the glacial to deglacial facies boundaries MIS 2/1 and MIS 6/5e. Shannon Wiener index values were relatively high (1.6 -2) at 40 cm and at 220-130 cm, within the glaciation and glacial (MIS 4-2) facies (Fig. 3). The interval from 350-320 cm is considered barren (Fig. 3), it contains only a few specimens of robust valve forms such as *Thalassiosira lentiginosa*, *Eucampia*

*antarctica* and *Actinocyclus actinochilus* (Table S4). This interval also contains pyritised shells, which were also found at 80-60 cm, during the Last Glacial Maximum (Fig. S1). The distribution of species (between 290-5 cm depth) is described below, in the order of species habitats (Table S1).

The most abundant species in the samples were those associated with open ocean environments, *Thalassiosira lentiginosa* (20-73% of the total counts); *Eucampia antarctica* (2-62 %), and *Fragilariopsis kerguelensis* (1-25%; Fig. 3). The highest abundance of *Thalassiosira lentiginosa* (>55%) occurred at 50-5 cm (interglacial and deglacial facies), at 150 cm (glaciation) and at 270-230 cm (interglacial). Highest abundance of *Eucampia antarctica* (18-62%) occurred at 210-60 cm (glacial and glaciation), at 290-280 cm (deglacial), and at 250 cm (interglacial). Maximum *Fragilariopsis kerguelensis* (17-25%) was found at 220-210 cm (glaciation) and at 50-40 cm (deglacial). Less abundant open ocean species (Fig. 3) included: *Thalassiosira oliveriana* (highest abundance of 4-8%) at 20-5 cm, 60 cm, 200-190 cm, 240-230 cm, and at 280 cm, within interglacial, deglacial and glacial facies; *Azpeitia tabularis* with abundance of 3-6% at 50-5 cm, and 260-250 cm, within the interglacial and deglacial facies; *Chaetoceros bulbosum* with 4%, at 40 cm, within the deglacial facies; *Chaetoceros dichaeta* with 4%, at 200 cm, within the glaciation facies; *Asteromphalus hyalinus* with 2-3%, at 150 cm, and 250 cm within the glacial facies. *Thalassiothrix group*, dominated by *Thalassiothrix antarctica* had a relative abundance of 3% at 40 cm (Fig. 3), and a high amount of broken valves relative to other samples, at 40 cm, and 270 cm (Fig. 5), although the 270 cm sample had 0% relative abundance (i.e., valve ends counted). Both intervals occur within the deglacial facies. *Thalassiosira oestrupii* had an abundance of 2-3%, at 50-30 cm (deglacial) and 240 cm (interglacial); *Asteromphalus parvulus* had 1.8% at 170 cm, and 2%, at 200 cm, within the glacial facies; and the *Rhizosolenia* group, dominated by *Rhizosolenia antennata* var. *semispina,* and less by *Rhizosolenia* antennata var. *antennata*, had a relative abundance of 1.8-1.9%, at 140 cm, 170 cm, and at 200 cm, within the glacial facies.

Open ocean- sea ice edge species (Table S1) comprised *Actinocyclus actinochilus* found from 7-13%, at 140-50 cm and 290-280 cm, within glacial and deglacial facies; and *Thalassiosira tumida*, found from 2-4% at 170-160 cm, 200-190 cm, within the glacial, and 250-230 cm, within the interglacial facies.

Sea ice proxies (Table S1) comprised the *Fragilariopsis* group species, *Eucampia* index and *Stellarima microtrias*. The *Fragilariopsis* group comprised a dominant species *Fragilariopsis obliquecostata* and much lower abundances of *F. sublinearis, F. linearis,* and *F. cylindrus. Fragilariopsis obliquecostata* is a species that lives in sea ice (Crosta et al. 2022; Garrison and Buck 1989; Armand et al. 2005). The *Fragilariopsis* group attained maximum abundances from 5-8% at 160-200 cm, within the glacial facies. The *Eucampia* index is elevated between 140-60 cm within the late MIS 4-2 glacial facies. The *Eucampia* index is not considered at intervals within 50-5 cm, and 270-220 cm, and at depths of 150 cm, 180 cm, and at 200 cm (Fig. 3), due to *Eucampia antarctica* counts being <100 valves per sample, considered too low to be statistically reliable. *Stellarima microtrias* was found >2% only at 270 cm, within the deglacial.

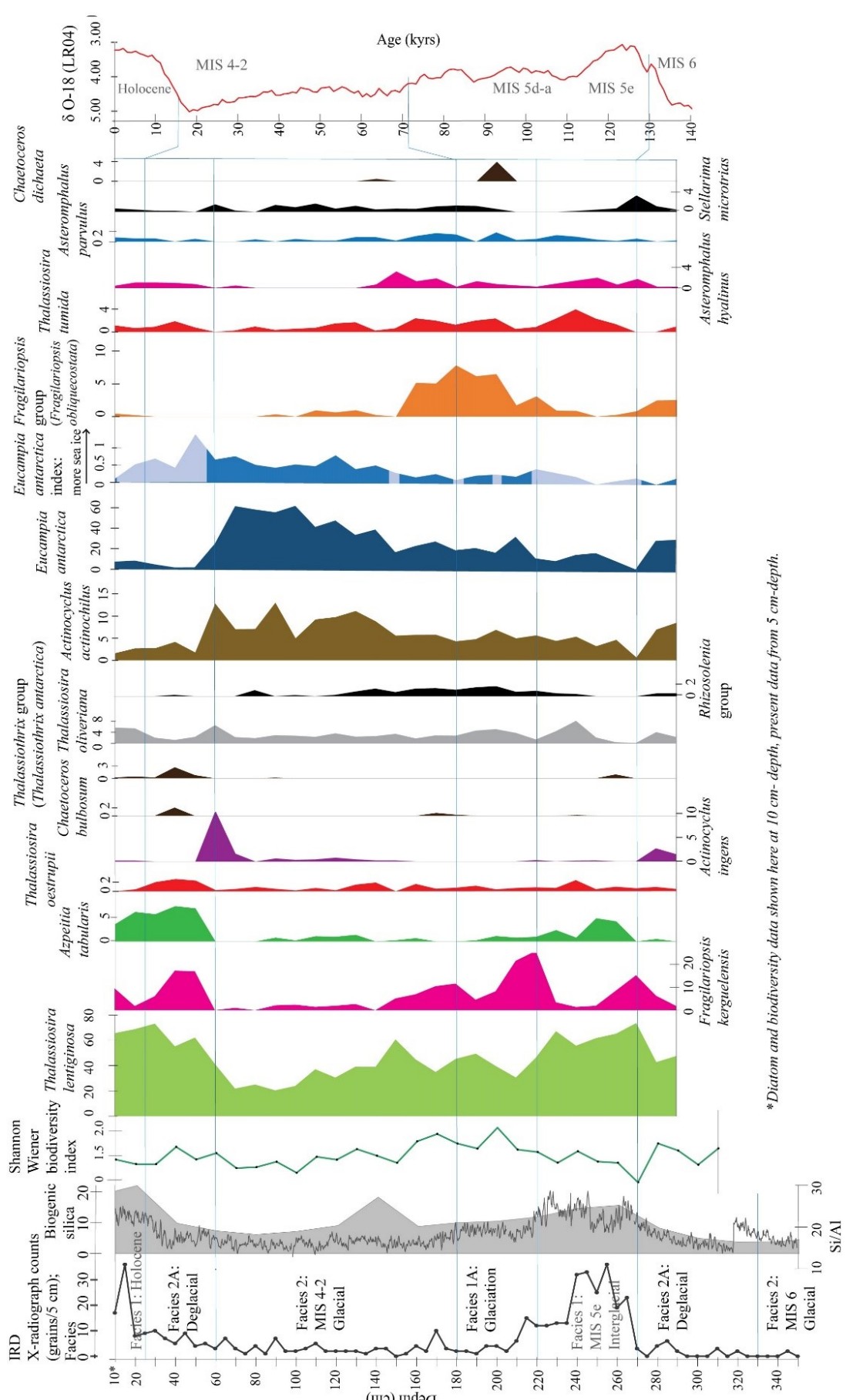

*Diatom and biodiversity data shown here at 10 cm- depth, present data from 5 cm-depth.

345

**Figure 3** Tan_44 distribution of main species, species groups and *Eucampia* index (Terminal/ Intercalary valve ratio). Results include IRD counts, facies interpretation (vertical lines), biogenic silica (%), Si/Al (XRF) and the Shannon-Wiener biodiversity index. The facies model is shown, in comparison to LR04 (Lisiecki & Raymo 2005). Species in black are not included in statistical analysis due to lower abundance (>1.8% in just one sample). These are *Rhizosolenia* group (mainly *Rhizosolenia antennata* var. *semispina* and *Rhizosolenia antennata* var. *antennata*), *Thalassiothrix* group (mainly *Thalassiothrix antarctica*), *Stellarima microtrias*, *Chaetoceros bulbosum,* and *Chaetoceros dichaeta*. *Eucampia* index was also not included in statistical analysis, its distribution in dark blue shows where total *Eucampia antarctica* counts are >100 valves per sample, while the light blue areas show samples with <100 counts.

### 3.4 Cluster and principal component analyses

Cluster analysis groups samples according to the similarity of the sample assemblages (Shi 1993). The groupings, illustrated by a dendrogram, can represent similar environments, and therefore aid in the reconstructions of paleoenvironments. Based on the relative abundance of diatom species, four clusters were identified (Fig. 4), with a dissimilarity index of 7%. The two largest groups, Cluster 1 and Cluster 2, correlate well with the interglacial and glacial facies, respectively (Fig. 4).

Cluster 1 includes samples from 5-50 cm, 150 cm, and 270-220 cm. This cluster, which contains open ocean species (Fig. 3), is associated with mainly the interglacial, deglacial and much less with glacial facies. Cluster 2 includes samples from 140-110 cm, 200-160 cm, and 60 cm. This cluster is mainly associated with the glacial facies, represented by sea ice and ice edge species (Fig. 3). Cluster 3 includes samples from 100-70 cm and is associated with the glacial facies. Cluster 4 is represented by only one sample, at 210 cm, within the glaciation facies.

The Q-mode PCA analysis identified three components that together explained 54% of sample variance (Table S2; Table 3). Component 1 (PC 1) explains 26% of the variance and contains contributor species associated with open ocean and sea ice edge environments. Species determining this component are *Thalassiosira lentiginosa*, *Actinocyclus actinochilus*, *Eucampia antarctica*, *Azpeitia tabularis*, and *Asteromphalus hyalinus.* Component 2 (PC 2) explains 16% of the variance and is associated with sea ice or the coastal Antarctic environment. These are the *Fragilariopsis* group (dominated by *Fragilariopsis obliquecostata*), *Asteromphalus parvulus*, and *Thalassiosira tumida*. Component 3 (PC 3) explains 12% of the variance and its contributor species are associated with open ocean environment. These are *Actinocyclus ingens*, *Actinocyclus actinochilus*, and *Thalassiosira oliveriana*.

R-mode PCA analysis identified three components, explaining 99% of the down core variance (Table S3; Fig. 5). The variance is mostly explained by PC 1 and PC 2. PC 1, the open ocean assemblage, explains 54% of the variance and shows high factor loadings at 40-5 cm and 270-150 cm. Both of these intervals coincide with Cluster 1, 2 and 4, in the interglacial, deglacial, glaciation and early MIS 4-2 glacial facies. PC 2, the sea ice, and ice-edge species assemblage (Table 3), explains 42% of the variance and shows high factor loadings at 140-70 cm, 170 cm, 210 cm, and 290-280 cm. These intervals coincide with Cluster 2 and 3, in the glacial facies and Cluster 4 in the glaciation facies (Fig. 5).

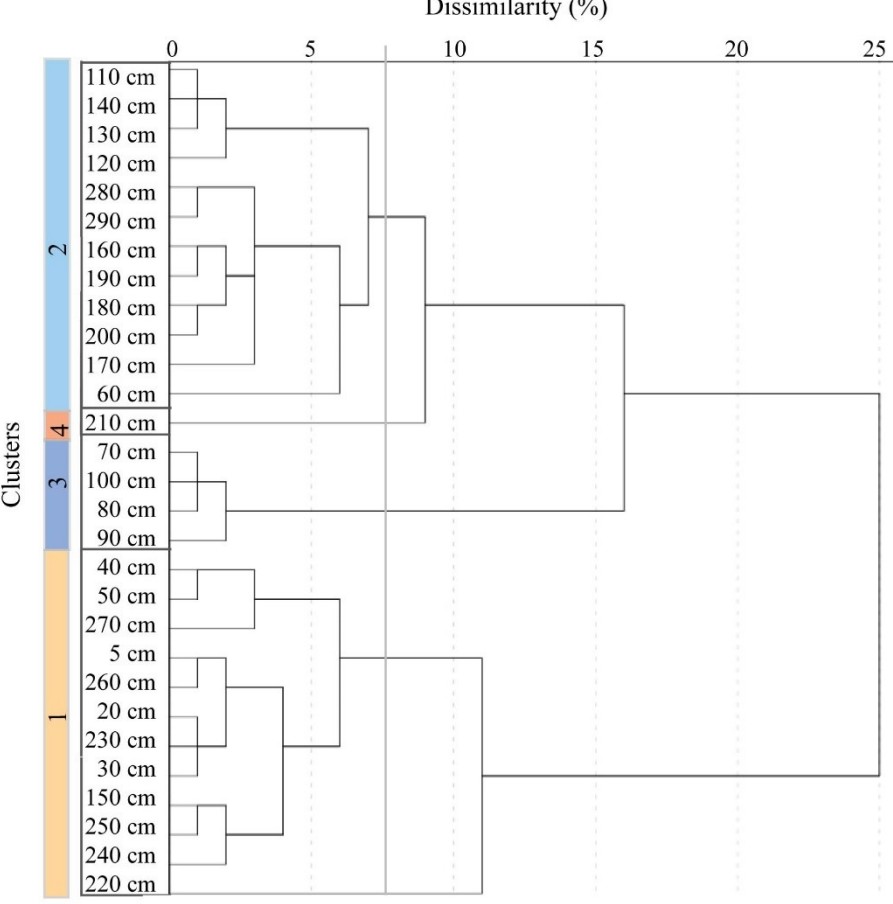

**Figure 4** Hierarchical cluster analysis dendrogram illustrating agglomeration of four clusters at dissimilarity of 7%.

390

**Table 3** Species assemblages (PC 1-PC 3) according to Q-mode principal component analysis (further information can be found in Table S2).

| Factor loadings | Assemblage | Environment |
|---|---|---|
| | P 1 | |
| >0.5/ >-0.5 | *Thalassiosira lentiginosa* | Open ocean |
| | *Eucampia antarctica* | |
| | *Azpeitia tabularis* | |
| | *Astermophalus hyalinus* | |
| | *Actinocyclus actinochilus* | Sea ice edge |
| | P 2 | |
| >0.5 | *Fragilariopsis* group* | Sea ice |
| | *Thalassiosira tumida* | Ice edge |
| | *Asteromphalus parvulus* | Coastal |
| | P 3 | |
| >0.5 | *Actinocyclus ingens* | Reworked |
| | *Actinocyclus actinochilus* | |
| | *Thalassiosira oliveriana* | |

\* mainly *F .obliquecostata*

## 3.5 Correlation between diatom assemblages and productivity

Correlation analysis shows a strong statistical relationship between PC 1 and PC 2 assemblages, and Si/Al and biogenic silica (Table 4). PC 1 assemblage shows a positive correlation to Si/Al ($r=0.63$), and to biogenic silica ($r=0.57$). PC 2 shows a negative correlation to Si/Al ($r=-0.62$) and to biogenic silica ($r=-0.54$). All correlations are statistically significant ($p < 0.001$).

**Table 4** Correlation (r) between each PC component and Si/Al and biogenic silica.

| | r value | |
| --- | --- | --- |
| Assemblage | Si/Al | Biogenic silca |
| PC 1 open ocean | 0.63* | 0.57* |
| PC 2 sea ice | -0.62* | -0.54* |

* statistically significant correlation ($p < 0.001$)

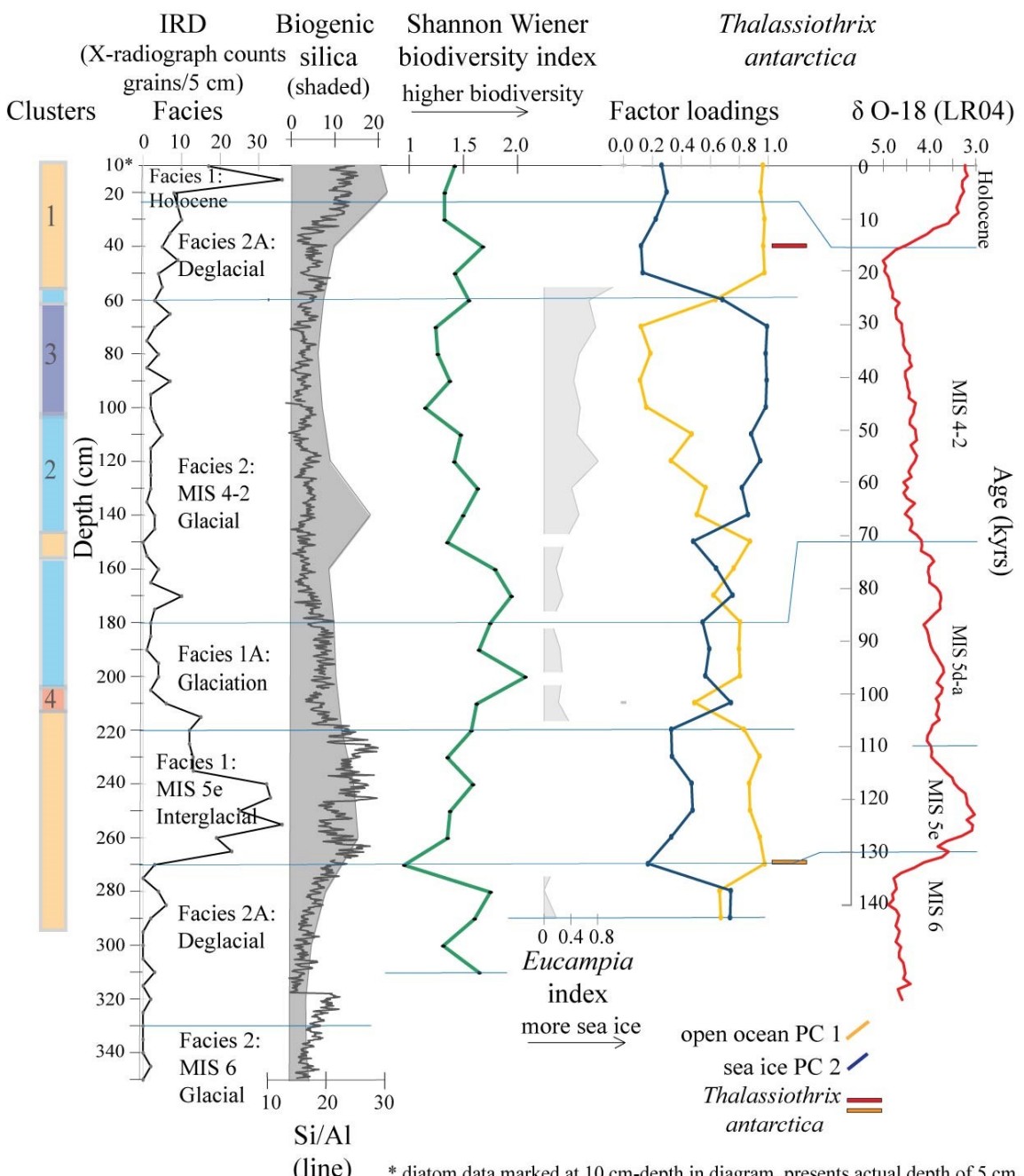

**Figure 5** Principal Component PC 1 and PC 2 factor loadings down core Tan_44, *Eucampia* index, *Thalassiothrix antarctica*, and Shannon-Wiener biodiversity index (green). Also included are cluster results, IRD counts, sediment facies (horizontal lines), biogenic silica, Si/Al and LR04 curve (Lisiecki & Raymo 2005). *Eucampia* index results are presented here only at depths where total *Eucampia antarctica* counts >100 valves per sample. *Thalassiothrix antarctica* (red and orange) show depths with abundant broken valves, where 270 cm sample (orange) comprised 0% relative abundance of counted valve ends.

**4 Discussion**

**4.1 Diatom assemblages, clusters, and sedimentary facies**

Principal component analysis distinguished 3 diatom assemblages. PC 1 and PC 2 assemblages incorporated most of the variance (42-54%), while PC 3 was much less influential (accounting for ~2% of total variance) (Fig. 5). The assemblages and their environmental interpretation are described below. Due to PC 3 contributing a minor amount of variance to the samples, it is defined here, but it's not used in the environmental interpretations. Separately from the statistics, the *Eucampia* index and the presence of lower abundances of

indicator species, in particular the *Thalassiothrix* group – which indicates high productivity, are further discussed.

**4.1.1 The open ocean assemblage (PC 1)**

The PC 1 assemblage comprises open ocean species, *Thalassiosira lentiginosa*, *Eucampia antarctica*, and *Asteromphalus hyalinus* (Johansen and Fryxell 1985; Garrison and Buck 1989; Medlin and Priddle 1990; Zielinski and Gersonde 1997); ice edge species, *Actinocyclus actinochilus* (Medlin and Priddle 1990; Ligowski, Godlewski and Lukowski 1992; Garrison and Buck 1989; Armand et al. 2005), and warmer water species, *Azpeitia tabularis* (Zielinski and Gersonde 1997; Romero et al. 2005). Therefore, the PC 1 assemblage is

interpreted to represent an open ocean environment, relatively warmer ocean, with a seasonal sea ice cover (Table 3) and high productivity, similar to the modern-day environment over the core site.

The composition of the PC 1 assemblage further suggests that selective species preservation, due to reworking by bottom currents and/or dissolution processes, had been active. The presence of a combination of robust

species, e.g., *Eucampia antarctica*, and *Actinocyclus actinochilus,* suggests that some level of reworking of sediments (Shemesh, Burckle, and Froelich, 1989; Taylor and McMinn 1997) or dissolution (Warnock and Scherer 2015) influenced the assemblage composition. These species have been found within assemblages considered to have been influenced by reworking offshore Cape Darnley in Prydz Bay (Taylor and McMinn 1997) and the continental slope of the Ross Sea (Truesdale and Kellogg 1979). Reworking is corroborated by

the knowledge that the site is currently influenced by the down slope flow of Adélie AABW and along slope currents, including the ASF (Fig. 1; Williams et al. 2008). Furthermore, the presence of unusual abundances of *Thalassiosira lentiginosa* (Fig. 3), a species usually associated with open ocean assemblages (Taylor and McMinn 1997; Truesdale and Kellogg 1979; Crosta et al. 2005) has been associated with dissolution (Shemesh, Burckle, and Froelich, 1989) further supports that some level of dissolution had affected the PC 1 assemblage

composition. Such high abundances of *T. lentiginosa* are not observed in modern sediments in the Adélie region (Leventer 1992), or elsewhere within the sea ice zone on the Antarctic margin (Zielinski and Gersonde 1997; Armand et al. 2005; Crosta et al. 2005). Despite the influence of reworking and dissolution, the PC 1 assemblage is still considered to be primarily autochthonous and dominated by *in-situ* deposition associated with open ocean, warmer water, and sea ice edge species. The presence of *Azpeitia tabularis,* and

*Asteromphalus hyalinus*, species not commonly associated with reworking or dissolution, further confirms this position.

### 4.1.2 The sea ice assemblage (PC 2)

The largest contributions to the PC 2 assemblage (Table 3) are from the *Fragilariopsis* group, comprising mainly *Fragilariopsis obliquecostata*, a species which currently lives in sea ice and at the sea ice edge (Ligowski, Godlewski & Lukowski 1992; Medlin & Priddle 1990; Moisan & Fryxell 1993). In Antarctic continental margin surface sediments from the Ross Sea, Weddell Sea and Prydz Bay, *Fragilariopsis obliquecostata* appears where sea ice cover is present >7 months per year (Armand et al. 2005). Other species

in PC 2 include the coastal species, *Asteromphalus parvulu*s (Kopczynska et al. 1986; Scott and Thomas 2005), and the open water/ sea ice edge species *Thalassiosira tumida* (Garrison & Buck 1989). Based on a combination of sea ice and sea ice edge/ coastal species, the PC 2 assemblage is interpreted as resulting from an environment proximal to the permanent sea ice edge with a long sea ice duration of >7 months, as currently observed in the Ross and Weddell Seas (Fetterer et al. 2017).


### 4.1.3 The reworked assemblage (PC 3)

The PC 3 assemblage comprises *Actinocyclus ingens*, *Actinocyclus actinochilus* and *Thalassiosira oliveriana*

(Table 3). *Actinocyclus ingens* is an extinct species, with LAD 0.43-0.5 Ma (Cody et al. 2008). *Thalassiosira oliveriana* is a species associated with open ocean environments (Medlin and Priddle 1990), while *Actinocyclus actinochilus* is associated with sea ice edge environments (Medlin and Priddle 1990; Garrison and Buck 1989). However, these species (including *A. ingens*) have robust valves that can survive transport by bottom currents (Shemesh, Burckle & Froelich 1989; Taylor and McMinn 1997; Truesdale and Kellogg 1977). PC 3 is therefore

interpreted as a reworked assemblage (allochthonous), transported from elsewhere by bottom water transport, with no *in situ* deposition, and hence no environmental signal. The reworking is supported by the presence of extinct *Actinocyclus ingens*, which is only found at 60 cm and 290-280 cm depth (Fig. 3). The 60 cm interval also contains pyritised shells (Fig. S1). Due to its very small influence on variability, the PC 3 assemblage is not considered further in the paleoenvironmental interpretation of Tan_44.


### 4.1.4 *Thalassiothrix antarctica* – a high productivity proxy

Aside from statistical analysis, the down core distribution of the *Thalassiothrix* group, of which *Thalassiothrix antarctica* is the most common species, are considered as environmental indicators (Fig. 3; Fig. 5).

*Thalassiothrix antarctica,* as well as the other two species which make up this group, *Thalassiosira lentiginosa* and *Trichotoxon reinboldii,* are open ocean species (Kopczynska 1998; Garrison and Buck 1989; Beans et al. 2009). *Thalassiothrix antarctica* and *Thalassiothrix longissima* are found in surface sediments, between coastal

Antarctica and the subtropical front (Zielinski and Gersonde 1997). While *Trixotoxon reinboldii* is associated with sediment from colder/ ice edge waters (Crosta et al. 2005), *Thalassiothrix antarctica* is also associated with diatom blooms that occur in modern coastal and shelf waters, such as in Prydz Bay (Ligowski 1983; Quilty et al. 1985) or in naturally fertilised areas, such as the Kerguelen Plateau (Rembauville et al. 2015). Zielinski and Gersonde (1997) consider *Thalassiothrix antarctica* from the Weddell Sea sediments to be indicators of high productivity. *Thalassiothrix antarctica* is not found in modern sediments off Adélie region (Leventer 1992). However, Beans et al. (2009) found that this species can sometimes be abundant in waters offshore Adélie Land. Based on the conclusions of Zielinski and Gersonde (1997) and Rembauville et al. (2015), the abundance of the *Thalassiothrix* group, represented largely by *Thalassiothrix antarctica* species, is indicative of a higher nutrient environment and higher productivity than in the present Adélie region, which we associate with increased upwelling of the Circumpolar Deep Water.

## 4.2 Palaeoecological interpretation

### 4.2.1 Interglacial (MIS 5e and Holocene)

The interglacial facies is associated with PC 1 (open ocean assemblage; Fig. 5; Table 3), and to a lesser extent with PC 2 (sea ice assemblage). The dominance of the PC 1 assemblage suggests that the Holocene and MIS 5e environments had seasonal sea ice, with open ocean during the summer and sea ice cover during winter, spring, and autumn, similar to the modern situation (Fig. 1). The inference of the seasonal presence of sea ice is strengthened with the moderate presence of PC 2 assemblage, which represents increased sea ice duration. PC 1 also provides evidence of reworked and a dissolution-affected assemblage. Indeed, strong bottom currents, such as AABW, and ASF, which sweep the continental slope offshore Adélie Land at present (Fig. 1; Williams et al. 2010), may have been active throughout the Holocene and MIS 5e. Furthermore, the reworking by bottom currents may have been stronger at times. Lastly, the PC 1 assemblage is associated with elevated productivity, which is supported by high Si/Al and biogenic silica, but lower biodiversity (Fig. 5). Low biodiversity is likely affected by poor preservation (dissolution and reworking), but potentially also reflects modern diatom blooms, which are typically of lower biodiversity (Beans et al. 2008). The close similarity of diatom assemblages between the two interglacial facies, MIS 5e and Holocene, suggests that ocean temperature, circulation, and seasonal sea ice duration in the Adélie region were similar during these two interglacial periods (Fig. 5).

### 4.2.2 Glaciation MIS 5d-5a (interglacial to glacial transition)

The glaciation facies shows clear evidence for an increase in the influence of sea-ice over the core site compared to the interglacial facies. The PC 2 assemblage starts to increase at this time, and the *Eucampia* index increases slightly (Fig. 5). *Fragilariopsis kerguelensis* reach maximum abundance early in the glaciation (Fig. 3), while the abundance of the *Fragilariopsis* group (dominated by *F. obliquecostata*) increases strongly throughout the

glaciation, reaching a maximum abundance at the transition to MIS 4-2 (Fig. 3). Finally, the *Rhizosolenia* group, of which the cold-water species *Rhizosolenia antennata* var. *semispina* (Armand et al. 2011; Fig. 3) dominate, are found throughout the glaciation facies. Together, the PC 2 assemblage, *Fragilariopsis* species, *Eucampia* index, and *Rhizosolenia antennata* var. *semispina* suggest that the glaciation facies was characterised by an

increase in the sea ice season, relative to the MIS 5e interglacial.

The productivity proxies, Si/Al, and biogenic silica are low, indicating a decrease in productivity throughout the glaciation (Fig. 5). However, the opposite is indicated by the continued presence of PC 1, which suggests high productivity.


The Shannon Wiener index suggests that the late glaciation was a time of relatively high biodiversity (Fig. 5) relative to the MIS 5e and Holocene interglacials. This may be due to a more diversified environment, that is, the increased sea ice season, and times of open water, may have produced a more diversified community. In the modern shelf environment, a greater diatom biodiversity is found near the Astrolabe Glacier, rather than near the

Mertz Glacier, where productivity is higher, yet dominated by fewer species (Beans et al. 2008). Thus, the biodiversity in the samples may reflect this natural variability seen in the diatom assemblages in the Adélie region today.

### 4.2.3 Glacial (MIS 4-2)


Diatom assemblages can be used to subdivide the MIS 4-2 glacial interval into early and late glacial stages (Fig. 5). The early glacial stage, comprising increased loadings of PC 1 (open ocean assemblage) and increasing loadings of PC 2 (sea ice assemblage), is similar to the glaciation, suggesting an initially prolonged sea ice season relative to the interglacial periods. The *Fragilariopsis* group is at its maximum in the early glacial stage

(Fig. 3). After this initial increase in PC 2, the assemblages align with high PC 1 only, suggesting temporary reversal of cooling, and an increase in productivity. Overall glacial productivity in this region was low with low Si/Al and biogenic silica. This is consistent with data from the broader Antarctic margin data (Bonn et al. 1998) and with the glaciation and deglacial facies. In the late glacial, from 160-100 cm, the assemblage displays a gradual increase in PC 2, suggesting an increase in the duration of the sea ice season (Fig. 5). After this, from

100-70 cm, the assemblages align with high PC 2, the sea ice assemblage, suggesting maximum duration of the sea ice season occurred towards the end of the late glacial stage (MIS 2). The increase in sea ice is further supported by the increase in the *Eucampia* index (Fig. 5).

The similarities in diatom assemblages between glaciation and early glacial stage suggests cooling of the ocean

started long before the onset of the glacial and then continued gradually until the maximum sea ice duration (and therefore, cooling) was reached, at the end of the last glacial (Fig. 5). This is consistent with gradual cooling reaching a maximum at the end of MIS 2, as seen in Antarctic ice cores (Jouzel et al. 1993) and Sea Surface Temperatures from global sediment cores (Kohfeld & Chase 2017), including records based on diatom assemblages from the Southern Ocean north of 56 °S (Crosta et al. 2004; Chadwick et al. 2022; Jones et al.

2022). The assemblage composition of the late glacial stage is suggestive of a long sea ice season duration, an environment which at present occurs in the Ross and Weddell Seas (Truesdale & Kellogg 1979; Zielinski & Gersonde 1997). This suggests that the permanent/ summer sea ice edge, during the late glacial stage, was closer to the core site than it is today, indicating that the core site was covered by near permanent sea ice during the peak glacial. However, the persistent presence of the *Thalassiosira lentiginosa* and PC 1, provides evidence that open ocean conditions existed over the Tan_44 site during part of the year.

**4.2.4 Deglacials (glacial to interglacial transitions): MIS 2 to Holocene and MIS 6 to MIS 5e**

The deglacial facies (between MIS 6 to MIS 5e and MIS 2 to Holocene) is generally associated with an increase in PC 1 (open ocean assemblage) and a decrease in PC 2 (sea ice assemblage) relative to the glacial (Fig. 5). The dominance of PC 1 suggests that there was high productivity throughout the deglacial, although the productivity proxies, biogenic silica, and Si/Al, are low. The minor influence of *Thalassiothrix antarctica* (at 40 cm; Fig. 5), relative to the glacial period, suggests an increase in CDW occurred after the decline of sea ice (at 60 cm, at end of the last glacial; Fig. 5) and prior to ice sheet retreat (at 15 cm, during the Holocene; Fig. 5). A similar sequence is observed within the MIS 6 to MIS 5e deglacial, but it is less clear. Here (at 270 cm; Fig. 5), the broken valves are abundant, but relative abundance is 0%, Tolotti et al. (2013), and Li et al. (2021), also suggest, based on the presence of diatom species, that increased CDW influx occurred during the last deglacial in the Ross Sea and offshore Enderby Land, respectively. Lastly, the start of the MIS 2/1 deglacial (and end of MIS 6/5 deglacial) is marked by a decrease in the PC 2 assemblage (Fig. 5), which suggests that the sea ice season duration declined rapidly at the end of the last glacial. This is consistent with the rapid sea ice retreat for the last two deglacial transitions from distal Southern Ocean (at 56°S) documented by Crosta et al. (2004) and Chadwick et al. (2022).

In conclusion, the duration of the sea ice season decreased relatively rapidly, and prior to CDW increase as evidenced by *Thalassiothrix antarctica* (Fig. 5), this in turn occurred before the onset of ice sheet retreat (as indicated by IRD; Fig. 5). However, a more detailed study of diatom assemblages is needed in order to determine the relative timing and a more detailed chronology of these deglacial changes.

**4.2.5 Diatom barren MIS 6 glacial and micropyrite in MIS 6 to MIS 5e deglacial**

The 350-320 cm interval is considered diatom barren (Table S4). This may be due to the original assemblage having been affected by dilution at the sea floor by turbidity currents (Kellogg & Truesdale 1979; Schrader et al. 1993; Escutia et al. 2003), or by a permanent sea ice cover, which reduced productivity allowing only reworked diatoms to be transported to the site by bottom currents (Table S4). The presence of micropyrite within the 320-300 cm section suggests low oxygen levels could have prevailed during the time, brought on by either fast sedimentation, such as turbidity currents (Presti et al. 2011), or by an extensive sea ice cover (Lucchi et al.

2007). Interestingly, pyrite is also found within the 80-60 cm section, which comprises an increase of the reworked assemblage (PC 3; Fig. 5) at the end of the MIS 4-2 glacial facies.

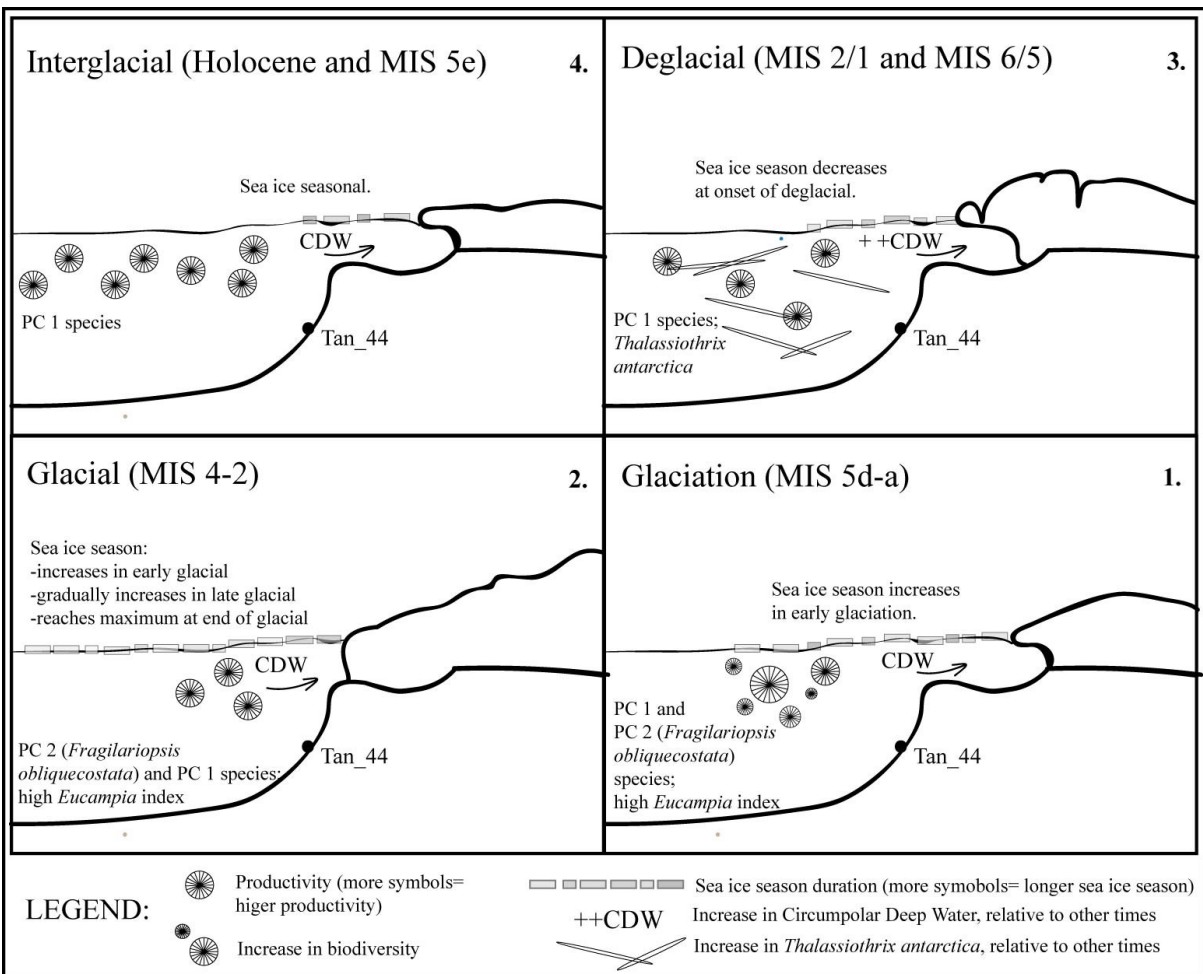

**Figure 6** Sea ice and productivity interpretation across the last glacial cycle (~140- ~5 ka) based on diatom assemblage variability. During Holocene and MIS 5e interglacial periods, the seasonal sea ice at the site is similar to present-day. Sea ice duration initially increases, then decreases in glaciation (MIS 5d-a); and this also occurs in the early MIS 4-2 glacial. The sea ice then, gradually increases during late MIS 4-2 glacial, reaching a maximum seasonal duration towards the end of the last glacial. During the deglacials (i.e., last two glacial to interglacial transitions) the influx rate of CDW increased relative to modern influx rates or other times, as suggested by the presence of the high nutrient species, *Thalassiothrix antarctica*. This study suggests the sea ice decreased at the end of the last glacial, and that the increase in CDW influx occurred after the sea ice season declined, during the last deglacial. Yet, all of this occurred before the onset of the last major ice sheet retreat.

## 5 Conclusion

Diatom assemblages in Tan_44 (64.5°S) varied on a glacial to interglacial timescale over the last 140 kyrs, and their composition reflects both *in situ* productivity, but is also influenced slightly by bottom current reworking processes. The diatom assemblages were predominantly influenced by changes in sea ice duration and changing ocean circulation over the core site. The following is a summary of conclusions reached in this study (Fig. 6):

- The PC 1 assemblage dominance in the interglacial facies suggests open ocean and seasonal sea ice environments during MIS 5e and Holocene, were similar to those of present day. The close correspondence of assemblages between the two interglacials suggests that both surface water conditions, related to sea surface temperature and sea ice duration were similar between the two interglacials.

- The unusually high dominance of *Thalassiosira lentiginosa* in the PC 1 assemblage in the interglacial facies suggests the assemblage was slightly affected by dissolution, and also by reworking by bottom currents.

- The duration of the sea ice season, as indicated by the presence of PC 2, started to increase during the early glaciation stage (MIS 5d-5a), and in the early MIS 4-2 glacial, and continued to increase throughout the late MIS 4-2 glacial, reaching a maximum extent towards the end of the glacial (MIS 2).

- However, the presence of both PC 2 and PC 1 assemblages during the MIS 4-2 glacial provides evidence that the summer sea ice edge was located further south than the core site, at 64.5°S in the Adélie region.

- The rapid decrease of the PC 2 sea ice assemblage suggests a decline in the sea ice season occurred at the end of the last glacial.

- *Thalassiothrix antarctica* was found to be abundant in both deglacials, suggesting an increase of high nutrient water. This influx is interpreted here as increase in upwelling of CDW in the Adélie region, during these times.

- Based on the diatom assemblages and the IRD increase in the Tan_44 core, the sea ice retreated prior to the increase in CDW upwelling during the last deglacial, which was then followed by an increase in IRD, indicating the retreat of the ice sheet.

- Biodiversity was highest during the glaciation stage and the start of the last glacial, and lowest during the late glacial and the interglacial periods. The higher biodiversity during glaciation and the start of the glacial could be the result of a more diversified environment relative to other periods.

This new diatom data set provides an understanding of the changes in sea ice proximal to East Antarctica over the last glacial cycle. It provides new insight into the extent of the summer sea ice in the last glacial, the winter sea ice extent during the last interglacial, and also suggests changes within seasonal sea ice with respect to the upwelling of CDW, during climate (glacial to interglacial) transitions. These are important parameters to constrain climate models to understand the importance of the influence of Antarctic sea ice on global climate over the last 140 kyr. Further marine sediment core data (increased resolution and more cores) is required to improve our understanding of the spatial and temporal changes in sea ice proximal to Antarctica.

**Data availability**

The data is available at PANGAEA: https://doi.org/10.1594/PANGAEA.946549

**Competing interest**

The authors declare there is no conflict of interest in relation to work presented in this study and in the Supplement.

## Acknowledgements

This research was supported under Australian Research Council's Special Research Initiative for Antarctic Gateway Partnership (Project ID SR140300001). We acknowledge National Institute of Water and Atmospheric Research (NIWA), Wellington, New Zealand and the crew and scientists from RV Tangaroa, from the 2013 voyage lead by Dr Mike Williams. This TAN1302 voyage was funded by New Zealand, Australia and French research agencies. The core was originally logged onboard ship by Dr Molly Patterson. Dr Taryn Noble assisted with u channel preparation at NIWA in Wellington, while the XRF scanning was completed by Patricia Gadd at ANSTO laboratories in Sydney. We further acknowledge Dr Nils Jansen who assisted with biogenic silica analyses at IMAS, Lisa Northcote, who assisted in grain size analysis, and Geraldine Jacobs and Alan William at the Australian Nuclear Science and Technology (ANSTO) who coordinated [14]C analysis. We also acknowledge the funding for 14C dating, provided by ANSTO Research Portal Proposal 10705.

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
