# Peer review of "Sea ice and productivity changes over the last glacial cycle in the Adélie Land region, East Antarctica, based on diatom assemblage variability"

_EGUsphere, 2022_

## Author Response (AR1)

**Comments to the author**:
Dear Dr. Pesjak,

Based on reviewers' comments and my own reading, I recommend reconsideration of your manuscript following major revision. Please provide a revised version along the lines you detailed in your response letters. Please also consider the editorial comments presented below:

Thank you. Please note that beside the major changes noted in the answers below, I have also focused the Discussion on PC 1 and PC 2, and eliminated PC 3 and PC 4 assemblages from down core interpretation. PC 3 is still discussed in the first part of Discussion, but it is concluded that due to its low influence on variability it is not used for paleoenvironmental interpretation of core.

Additional information about diatoms are needed in section 2.7. More specifically :
- The Fragilariopsis group is not explain (or I missed the information). Which species are included ?
Answer: This is already explained in the Results: Section 3.1 (line 467)
"Sea ice proxies (Table S1) comprised the *Fragilariopsis* group species, *Eucampia* index and *Stellarima microtrias*. The *Fragilariopsis* group comprised a dominant species *Fragilariopsis obliquecostata* and much lower abundances of *F. sublinearis, F. linearis, F. cylindrus* and *F. rhombica*. *Fragilariopsis obliquecostata* is a species that lives in sea ice (Crosta et al. 2022; Garrison and Buck 1989; Armand et al. 2005). The *Fragilariopsis* group attained maximum abundances from 5-8% at 160-200 cm, within the glacial facies.."

However, I added now as suggested, in Methods (line 279):
"Some species were grouped together due to morphology and habitat indicators, these groups are the *Fragilariopsis* group, comprising *F. obliquecostata, F. sublinearis, F. cylindrus,* and *F. rhombica*; the Thalassiothrix group, comprising .."*;

And line 289:
"The relative abundance of each species (or group) was expressed as the number of valves of that species divided by the total valve count (expressed as %). Species or species groups with >1.8% in at least two samples were included in statistical analysis, except in cases of the *Fragilariopsis* group, which apart from *Fragilariopsis obliquecostata* (present at >1.8% in at least 2 samples) also included much rarer sea ice species (*F. sublinearis, F. linearis, F. cylindrus* and *F. rhombica)*".

- There is no mention on how many specimens are calculated the T/I values. Generally, more than 100 valves need to be identified to produce robust ratio values. Here, E. antarcticta represents 2-5% of the diatom assemblages during the deglacation and Holocene. As ~400 diatom valves were counted, this suggests that the T/I was calculated on 8-20 E. antarctica valves. Results are therefore not statistically significant for this period, which is of prime importance for the paleoceanographic interpretation. This needs to be discussed.

Answer: Yes, the E. antarctica counts are very low for 50-30 cm and at 270 cm, and they are <100 counts per sample, within 50-5 cm, at 150 cm, at 180 cm, at 200 cm, and from 270-220 cm. This has now been added to manuscript, and *Eucampia* index for these intervals are now not taken into final interpretation.

This is now discussed in Methods, line 309:
"The *Eucampia* index was also as an indicator of sea ice presence (Fryxell et al. 1991). It represents the ratio of the number of terminal valves to the number of intercalary valves of *Eucampia antarctica* species, and its increase is associated with more sea ice in the environment. In the open ocean the *Eucampia antarctica* species grow in longer chains, while in sea ice waters they grow in shorter chains (Fryxell 1991). The chains comprise intercalary valves in the middle, and terminal valves at the ends, and therefore, the more terminal valves, the more sea ice (Fryxell 1991: Kaczmarska et al. 1993). The *Eucampia* index was only calculated where the total *Eucampia antarctica* count was 100 valves and above."

And in Results – line 473:

"..160-200 cm, within the glacial facies. The *Eucampia* index is elevated between 140-60 cm within the late MIS 4-2 glacial facies. The *Eucampia* index is not considered at intervals within 50-5 cm, and 270-220 cm, and at depths of 150 cm, 180 cm, and at 200 cm (Fig. 3), due to *Eucampia antarctica* counts being <100 valves per sample, considered too low to be statistically reliable."

Also the unreliable results are whited out in Figures 3 and erased from final, Figure 5.

-In section 2.8, please precise that statistical analyses are performed on diatom relative abundances.
Answer: This has now been added in line 319:
**"2.7 Statistical analyses: cluster analysis and principal component analysis**

The relative diatom abundance data set was analysed using a hierarchical cluster analysis and principal component analysis (PCA)...."

-I wonder whether all diatom species presented in Figure 3 can be included in the statistical analyses. Indeed, few species are present in low abundances and in only few samples. More specifically :
- C. dichaeta is present in 2 samples with only one above the 2% threshold.

-Answer: I had used the threshold of 2% in at least one sample as Taylor and McMinn (1997). However, I reconsidered my data, and decided to eliminate the species which occur in only one sample, above 2%.  So I have now
set the parameters at: >1.8% in at least 2 samples for inclusion in the statistical analysis. This eliminated *C. dichaeta*, but also *C. bulbosum, Stellarima microtrias*, and *Thalassiothrix* group, from statistical analysis.

-These species are now eliminated from statistics (further explained below) but are still included in results, and in Fig. 3 (now coloured in black) due to environmental indications.
-Both are now explained in Methods, line 286:
"The relative abundance of each species (or group) was expressed as the number of valves of that species divided by the total valve count (expressed as %). Species or species groups with >1.8% in at least two samples were included in statistical analysis, except in case of the *Fragilariopsis* group, which apart from *Fragilariopsis obliquecostata* (present at >1.8% in at least 2 samples) also included much rarer sea ice species (*F. sublinearis, F. linearis, F. cylindrus* and *F. rhombica)*.

Species or species groups present at >1.8% in at least 1 sample, and thus excluded from statistics, were included in results and discussion due to their environmental indications (Table S1). These species include .."

- Rhizosolenia gp is never above 2%.
Answer: Yes, this is true, *Rhizosolenia* group is at 1.9% at 140 cm, it is also at 1.8 % at 170 cm and 200 cm. Initially I had included *Rhizosolenia* group in the statistics, because I had rounded up 1.9% to 2% (from 140 cm) so it fits the statistics limit at the time. Now, according to the newly set parameters, >1.8% in at least two samples, *Rhizosolenia* group can be fit into the statistics, but I had eliminated it anyway because it doesn't have a single species that fulfills the criteria (as I present with *Fragilariopsis obliquecostata* in *Fragilariopsis* group), but only the sum of all species make the criteria. Also, the presence of *Rhizosolenia* group doesn't influence the outcome of the statistics, however, interestingly it pairs with the sea ice group, which is good, because *R styliformis* makes up the largest part of the group.
I still include this group in the down core distribution (line 293), I explain why it is not in the statistics ("the sum of both species was up to 1.2-1.6%", line 297), and I also describe which species are within the group (line 282).

Line 282:
"Some species were grouped together due to morphology and habitat indicators, these groups are the *Fragilariopsis* group, comprising *F. obliquecostata*, *F. sublinearis*, *F. cylindrus*, and *F. rhombica*; the *Thalassiothrix* group, comprising *Thalassiothrix antarctica, Thalassiothrix longissima* and *Trichotoxon reinboldii, and the Rhizosolenia* group, comprising *Rhizosolenia styliformis, Rhizosolenia (twin process) antennata, R. antennata, R. hebetata, R.setigera, R. polydactyla, Rhizosolenia sp., and Proboscia intermis*."

Line 293 and line 297:
"Species or species groups present at >1.8% in at least 1 sample, and thus excluded from statistics, were included in results and discussion due to their environmental indications (Table S1). These species include the *Thalassiothrix antarctica* group, represented mainly by *Thalassiothrix antarctica*, and the *Rhizosolenia* species group, presented mainly by *Rhizosolenia styliformis* and *Rhizosolenia* (twin process) *antennata* (the sum of both species was up to 1.2-1.6% in three samples)."

-I am also very surprised to see that this gp is dominated by R. styliformis, which is a sea-ice related diatom present on the shelf (Ligowski's papers) but generally absent from offshore waters (spuriously named if present ; Armand and Zielinski, 2001). How sure are you about your identification?

Answer: *Rhizosolenia styliformis* was determined using Armand and Zielinski (2001) images.

- Thalassiothrix gp present in 5-6 samples with only one above the 2% threshold.

Answer: Yes, Thalassiothrix gp is >2% (2.7%) at 40 cm, and <2% in other 5 samples. Its also present at 270 cm, but at 0% counted – however the occurrence of broken valves is very similar to 40 cm (where valves are at counted at 2.7%).

This group is now eliminated from statistics, but still discussed and included in Fig. 3, and in Fig.5 (newly added)- as part of important results. The methods and results include mention of 270 cm sample, as it is at 40 cm, relative to all other samples, as this is considered a significant environmental signal for the entire paper. All other samples, beside 40 cm and 270 cm, which happen to be deglacial facies, contain almost unnoticeable broken valves of the *Thalassiothrix* group. This is graphically noted in Fig. 5 (and in Fig. 5 caption), and further explained in Methods, Results and Discussion, noted below:

This is now mentioned in Methods, line 297

" *Thalassiothrix antarctica* group, represented mainly by *Thalassiothrix antarctica*, and the *Rhizosolenia* species group, presented mainly by *Rhizosolenia styliformis* and *Rhizosolenia* (twin process) *antennata* (the sum of both species was up to 1.2-1.6% in three samples). The *Thalassiothrix* group is also discussed where there is a significant increase in broken valves, yet the relative abundance (i.e., counted valve ends) is 0%.".

This is now highlighted in Results, line 448:

*"…… Thalassiothrix group*, dominated by *Thalassiothrix antarctica* had a relative abundance of 3% at 40 cm (Fig. 3), and a high amount of broken valves relative to other samples, at 40 cm, and 270 cm (Fig. 5), although the 270 cm sample had 0% relative abundance (i.e., valve ends counted). Both intervals occur within the deglacial facies;;;.."

I reworked the discussion, slightly changing Section 4.1.4- line 650 (copied below) which was former high productivity PC 4 assemblage, and also, Section 4.2.4 deglacial- line 788 (copied below). *Thalassiothrix* valve illustrations are now added to the final sketch on paleoenvironment for the last two deglacials  Fig. 6.

Line 654:

**4.1.4 Thalassiothrix antarctica – a high productivity proxy**

"Aside from statistical analysis, the down core distribution of the *Thalassiothrix* group, of which *Thalassiothrix antarctica* is the most common species, are considered as environmental indicators (Fig. 3; Fig. 5). *Thalassiothrix antarctica,* as well as the other two species which make up this group, *Thalassiosira lentiginosa* and *Trichotoxon reinboldii,* are open ocean species (Kopczynska 1998; Garrison and Buck 1989; Beans et al. 2009). *Thalassiothrix antarctica* and *Thalassiothrix longissima* are found in surface sediments, between coastal Antarctica and the subtropical front (Zielinski and Gersonde 1997)….";

"...The minor influence of *Thalassiothrix antarctica* (at 40 cm; Fig. 5), relative to the glacial period, suggests an increase in CDW occurred after the decline of sea ice (at 60 cm, at end of the last glacial; Fig. 5) and prior to ice sheet retreat (at 15 cm, during the Holocene; Fig. 5). A similar sequence is observed within the MIS 6 to MIS 5e deglacial, but it is less clear. Here (at 270 cm; Fig. 5), the broken valves are abundant, but relative abundance is 0%, Tolotti et al. (2013), and Li et al. (2021), also suggest.."

- C. bulbosum present in 3-4 samples and never above 2%.
Answer: *C. bulbosum* is at 2.1% at 40 cm. This species is now eliminated from statistical analysis, according to new parameters set. But I have kept it in Results, due to environmental indications- as answered above.

- A. ingens present in only 4-5 with 2 samples above the 2% threshold.
Answer: Yes, *A. ingens* is present at >2 % in two samples only, at 11% at 60 cm, and 2.9% at 280 cm.
Due to the new parameters, I set (>1.8% in at least 2 samples), I have included *A ingens* in the statistics. I think that this species is a representative species to be used in the statistics. It is interpreted as part of a reworked assemblage, such assemblages have been mentioned by Kellogg and Truesdale (1979), Taylor and McMinn (1997)/ Taylor and McMinn (2001).

☐ Please try to replay statistical analyses with only abundant enough (and thus truly representative) species.
Answer: I have run the statistics again exploring different scenarios using $\log_{10}$ (x+1), as suggested, and different inputs:  >2% relative abundance in at least 1 sample (therefore including the 'low % species'), or >1.8% in at least 2 samples, and with or without *Eucampia* index. By setting the limit >1.8% in at least two samples, has eliminated most of the species suggested (*C dichaeta, C bulbosum, Stellarima microtrias, Thalassiothrix* group).

I have found using $\log_{10}$ (x+1) and setting the new limit (>1.8% in at least two samples) thus eliminating the rarer species mentioned above, produces the same results as before. However, I found that including *Eucampia* index when using the log (x+1) doesn't clearly separate open ocean and sea ice species. I also decided not to use the *Eucampia* index in statistics in general, as in some intervals of core it is not statistically reliable input, due to low *Eucampia* antarctica counts.

Therefore, I have decided to exclude the rarer species, and exclude the Eucampia index, but I decided, to keep the log (x+1) equation (as per, Taylor and McMinn 1997).

The results, figures and discussion are now amended, accordingly to the new parameters (Fig. 3; Fig 4; Fig. 5; Table 3, Table 4), and in the Supplement, Table S2

and Table S3. Note that this time, the PC 4 is completely eliminated, and only PC 1 and PC 2 are discussed in environmental interpretation.

This is noted in Methods, line 326:

"The factor variance used to extract the number of components for Q-mode analysis was established at ≥12 % variance. The factor variance used to extract the number of components for R-mode analysis was established at ≥42 %. Factor...."

In Results line 516:

"The Q-mode PCA analysis identified three components that together explained 54% of sample variance (Table S2; Table 3). Component 1 (PC 1) explains 26% of the variance and contains contributor species associated with open ocean and sea ice edge environments. Species determining this component are *Thalassiosira lentiginosa*, *Actinocyclus actinochilus*, *Eucampia antarctica*, *Azpeitia tabularis*, and *Asteromphalus hyalinus*. Component 2 (PC 2) explains 16% of the variance and is associated with sea ice or the coastal Antarctic environment. These are the *Fragilariopsis* group (dominated by *Fragilariopsis obliquecostata*), *Asteromphalus parvulus*, and *Thalassiosira tumida*. Component 3 (PC 3) explains 12% of the variance and its contributor species are associated with open ocean environment.. …"

And in Results, line 530:

"R-mode PCA analysis identified three components, explaining 99% of the down core variance (Table S3; Fig. 5). The variance is mostly explained by PC 1 and PC 2. PC 1, the open ocean assemblage, explains 54%.."

☐ I wonder whether it would be better to use log(%+1) as input data for the PCA to increase the representativity of low abundant species that may be of paleoceanographic importance.

Answer: This change is now addressed (line 320):

"**2.7 Statistical analyses: cluster analysis and principal component analysis**

The relative diatom abundance data set was analysed using a hierarchical cluster analysis and principal component analysis (PCA), in Statistical Package for Social Sciences (SPSS) software package. For these analyses the relative abundance data was logarithmically transformed using the equation: Abundance = $\log_{10} (x+1)$, where x= relative abundance (%), (Taylor, McMinn & Franklin 1997). Cluster analysis (Burckle 1984; Truesdale & Kellogg 1979) involved.."

Most of diatom species preserved in TAN44 are highly silicified diatoms. I wonder whether dissolution and mechanical break-up might not have biased the original signals and, therefore, the paleoceanographic interpretations. Any proof that dissolution did not strongly impact the diatom assemblages: presence of small and big diatoms, areolation well preserved, etc...

Answer: In general dissolution is not so evident on diatom valves, they are well preserved, areolation is well preserved and variation in size of diatoms is present- for example this is evident in smaller and bigger *T. lentiginosa*, larger *Coscinodiscus*, and larger *T. tumida*.

This is mentioned in line 396:

"**3.3 Species distribution, biodiversity, and abundance**

All samples contained well-preserved diatom assemblages with little evidence of dissolution, such as frustule thinning..."

In the Discussion of assemblage PC 1, line 597, dissolution (and reworking) is suggested to influence content of the assemblage, *to an extent*, with regard to interpreting the content of PC 1 and the unusual dominance of *Thalassiosira lentiginosa*:

"The composition of the PC 1 assemblage further suggests that selective species preservation, due to reworking by bottom currents and/or dissolution processes, had been active. The presence of a combination of robust species, e.g., *Eucampia antarctica*, and *Actinocyclus actinochilus,* suggests that some level of reworking of sediments influenced the assemblage composition (Shemesh, Burckle, and Froelich, 1989; Taylor and McMinn 1997). These species have been found within assemblages considered to have been influenced by reworking off Cape Darnley in Prydz Bay (Taylor and McMinn 1997) and the continental slope of the Ross Sea (Truesdale and Kellogg 1979). Reworking is corroborated by the knowledge that the site is currently influenced by the down slope flow of Adélie AABW and along slope currents, including the ASF (Fig. 1; Williams et al. 2008). Furthermore, the presence of unusual abundances of *Thalassiosira lentiginosa* (Fig. 3), a species usually associated with open ocean assemblages (Taylor and McMinn 1997; Truesdale and Kellogg 1979; Crosta et al. 2005) has been associated with dissolution (Shemesh, Burckle, and Froelich, 1989) suggesting that there is some level of dissolution affecting the PC 1 assemblage composition. Such high abundances of *T. lentiginosa* are not observed in modern sediments in the Adélie region (Leventer 1992), or elsewhere within the sea ice zone on the Antarctic margin (Zielinski and Gersonde 1997; Armand et al. 2005; Crosta et al. 2005). Despite the influence of reworking and dissolution, the PC 1 assemblage is still considered to be primarily autochthonous and dominated by *in-situ* deposition associated with open ocean, warmer water, and sea ice edge species. The presence of *Azpeitia tabularis,* and *Asteromphalus hyalinus*, species not commonly associated with reworking or dissolution, further confirms this position."

-Figure 3: Please remove the diatom counts / slide (and in subsequent figures too).
Answer: Removed from figures 2, 3 and 5 as suggested.

-Please consider removing « non-representative species » (the ones mentioned above).
Answer: Yes, this is now completed as described above, except *A. ingens*, and this is explained.

-Add MIS labels alongside the d18O LR04 record for better visualisation.
Answer: Added to Fig. 2, Fig. 3, and Fig. 5 as suggested.

-Figure 5 : The grounding line during the glacial was probably closer to the shelf break as no biogenic sediments are found on the shelf before 12 kyrs BP (Mackintosh et al., 2014). In this vein, Bentley et al. (2014) inferred the GL to be at the shelf break. No presence of sub-ice shelf cavity during this period allowing intrusion of CPDW on the shelf, leading to the hypothesis of open ocean AABW formation during this period (Paillard et al., 2004 ; Bouttes et al., 2010).

Answer: I think this question refers to the sketch in Figure 6. In my thesis (Pesjak 2022) I suggest based on evidence from other slope cores, that the grounding line in MIS 4-2 was not to the shelf edge over the banks, but that it was within the trough (based on the one shelf core) in the Adelie region.
As suggested here, I have extended this grounding line now in Fig. 6 to be closer to the shelf edge.

All the best.

Xavier Crosta
01 Dec 2022
**Final response**
Details
04 Oct 2022
**Discussion started, expected end 29 Nov 2022**
Minimum number of referee reports required: 2

Nominated Referee
nominated 04 Oct 2022, nomination terminated by editor
Referee #2: Chadwick, Matthew  m.chadwick@cornwall-insight.com
nominated 04 Oct 2022, accepted 07 Oct 2022, report 03 Nov 2022 Report #2
Anonymous Referee #1
nominated 04 Oct 2022, accepted 07 Oct 2022, report 30 Oct 2022 Report #1

**Initial submission**

04 Oct 2022
**Published**
Preprint
03 Oct 2022
**Editor initial decision: Start review and discussion**
by Xavier Crosta
03 Oct 2022
**Editor found**
Xavier Crosta agreed to serve as editor

Referee 1

General Comments:

I really enjoyed reading this study, which presents a paleoenvironmental interpretation of a sediment core recovered from the well north of the Adelie Land continental shelf, within the seasonal sea ice zone. The authors correctly identify a critical gap in our ability to reconstruct paleoceanographic conditions beyond the last deglaciation around the Antarctic margin, due to glacial advances across the shelf that restrict most sediment records to this limited time frame. This means, that to go farther back in time, yet, be relatively proximal to the continent, we must work on cores from the slope and rise, and then to the proximal deep sea. This study does exactly that, working with a core from farther offshore, on the slope, in a water depth of >3000 m. TAN 1302-44, a 3.5 meter, goes back to MIS6, and allows a reconstruction of glacial, deglacial and interglacial progression at this site, using a multi-proxy data set that relies heavily on the diatom assemblage data, and a chronology that is suggested based mostly on matching the Si/Al ratio to the global benthic d18O stack. Radiocarbon dates near the top of the core are also utilized, but they are limited to < the upper 50 cm. While this reduces the robustness of the age model, I also recognize that this is a problem for so many Southern Ocean cores, with an absence of foraminifera that could be used to develop a stable isotope record and hence, a more robust chronology. Overall, the authors do a very good job interpreting the diatom data, along with other proxies, and they provide a strong evaluation of changes in paleo sea ice extent and paleo-productivity over time. Their interpretation of the diatom assemblages is good, and of course, the statistical approach is appropriate, but here, perhaps add in more species-specific commentary – I suggest this below as well, for example when describing the oceanographic conditions suggested by F. obliquecostata and also for Thalassiothrix. Regardless of the principal components, I always go back to the species data! In summary, a strong paper that I recommend for publication; specific comments and questions are listed below; these are intended to add to the depth of their already strong interpretation.

Thank you for your kind comments and your work in bringing suggestions to this manuscript.

Specific Comments:

1. Use of the Eucampia antarctica terminal valve/intercalary valve ratio is appropriate here, as a way to estimate changes in winter sea ice extent – I suggest a more complete explanation of this ratio (Define it once, and then you can call it the Eucampia index, as done by others), and perhaps an interpretation of this in Figure 3, with an arrow indicating more sea ice to the right, and in Table 2 (what about the ratio – higher or lower? Be specific), line 337.

Answer: Index is introduced and defined in text (line 266 ); Figure 3 caption, and in Table S1 (Supplement). The index is corrected instead of terminal/intercalary ratio in Table 2, Figure 3 and in Table S3. Higher index is pointed out in Fig 3 with arrow showing more sea ice, and better defined in Table 2, as suggested. Index is also written in abstract (line 23).

Answer 12th January: the index is now defined in line 304.

2. And the authors correctly point out that in some of the intervals, the number of Eucampia counted are simply too low to have a statistically reliable number – in general this might be any time you've counted fewer than 100 specimens that could be identified as either terminal or intercalary, as many times that determination is not possible.

Answer: I agree.

Answer 12th January: This is now incorporated as per editor comments, line 309.

3. Also, I wondered which variety of Eucampia was present, var. antarctica or var. recta – or a mixture of the two?

Answer: This distinction wasn't made. It is likely that it is a mixture of the two, but that could also depend on the interval.

4. Third paragraph of introduction – perhaps slightly re-frame this to compare the utility and challenges of shelf versus slope/rise versus deep-sea records.

Answer: Re-framed paragraph (line 76-84).

Answer 12th January: Line 84.

5. What kind of core? (piston core?)

Answer: Added: gravity corer with a 2-tonne head (line 105).

Answer 12th January: Line 116.

6. Table 1: In the supplement you explain where the "% microfossil" estimates come from – but since I had a question about this as I read, I suggest that the explanation come in the main text as opposed to the supplement. With the diatom estimates, given that you are working with samples that you sieved, and that you made these estimates on the sand fraction (>63 microns), I am not sure how reliable this number is, even as an estimate. Bottom line, this methodological information should be up front, if you decide to retain the estimates in your paper, since this estimate doesn't include so many diatoms, which are mostly silt-sized. I don't have a recommendation either way.

Answer: Diatom estimates are included in main text now (line 150). I have not deleted them as they strengthen biogenic silica, Si/Al and IRD data.

Answer 12th January: This is now deleted from the manuscript and supplement all together, and Fig. S1.

7. Diatom counts: I am very comfortable with the diatom assemblage data, and roughly, but less so, the diatom counts. The diatom counts are useful in terms of evaluating if samples are

diatom-rich or very diatom-poor and the bSi data provide a quantitative comparison. But absolute abundance data might be helpful here (or in future work). I am not really sure why the diatom counts per slide are presented in figure 2 – since this is non-quantitative. The authors state this on line 199 – the qualitative nature of the data. In the future I suggest using a different technique to make quantitative slides, for example, perhaps adopting the method described by Scherer (1994) [Scherer, R. (1994). A new method for the determination of absolute abundance of diatoms and other silt-sized sedimentary particles. Journal of Paleolimnology, 12, 171–179. https://doi.org/10.1007/BF00678093] and revised by Warnock and Scherer (2014)[/ [Warnock, J. P. and R. P. Scherer (2014), A revised method for determining the absolute abundance of diatoms, J. Paleolimnol., doi:10.1007/s10933-014-9808-0.]

Answer: I agree, thank you for the suggestions. I have removed diatom counts per slide in Fig. 2 and Fig 3. Instead, I include IRD counts, as per Referee 2 suggestion. Following this, discussion on diatom count results was removed (line 678-682).

Answer January 12[th]: This erased discussion is now – line 811.

8.  Chronology – as noted in my comments above, the chronology is based on several radiocarbon dates in the upper 50 cm and comparison of the Si/Al data to the LR04 stack. Given the lack of foraminifera and the limits for radiocarbon dating, I think the authors have done what they can. I wondered if they are able to look carefully at the MIS6/5e boundary to see if samples from MIS6 have any Rouxia leventerae – a good biostratigraphic marker. This may not be possible, given the scarcity of diatoms in the MIS6 section. Any evidence for MIS3, which does show up in Sabrina Slope piston cores (Holder et al., 2020)? Perhaps looking carefully and at higher resolution around 140 cm, where there is an increase in bSi might reveal an indication of MIS3? It's a possibility. Perhaps as well, one deeper radiocarbon date? Also, note that the text, line 169 indicates 2 radiocarbon dates, but figure 3 shows 3 dates, and Table S2 has 4 dates listed. I suggest including the radiocarbon data table in the main paper, not in the supplement.

Answer: *Rouxia leventerae* wasn't identified in any of the slides analysed (this is now added to text; line 223) and all of the slides were carefully analysed, even the barren slides. However, additional analysis of the deeper core, older MIS 6, and MIS 7, may provide some answer to this question in the future.

Answer 12[th] January 2023: That R. leventerae isn't found is now mentioned in line 370.

I agree that a more detailed/ higher resolution analysis of diatom assemblages may help in distinguishing age and paleoenvironments, such as perhaps determining MIS 3. At this stage there is too little evidence for MIS 3, biogenic silica would need to be analysed in higher resolution also.

A deeper radiocarbon date isn't possible because in general the radiocarbon dates become unreliable at depth, in this case beyond 25 cm, which was seen in other 2 sediment cores in the area at similar depths (Pesjak 2022, thesis). The ages suggest a sedimentation rate which is too high. This problematic happens during the glacial as the sedimentation processes involve a lot more terrigenous matter influx, relative to biogenic.

I have brought the radiocarbon table into the manuscript now. I added an explanation in the Age Model section (line 214-217) explaining the reason why the two deeper dates were excluded. Figure 2, and Fig S1 show all 4 dates, as they are in original Table S2. Fig. 3 shows no dates.

Answer January 12[th]: The deeper radiocarbon dates are included in Table 2, in manuscript, and the problems with these dates are explained now in line 372, in Results.

9. Section 3.2 is overly long and detailed. I suggest paring this section down to highlight specifics that are critical to the interpretation The details can be found in the supplementary material data table.

Answer: This section (now named as Section 3.1) is now rewritten to highlight interpretation as suggested by the referee, and it is also simplified (line 315).

Answer 12[th] January: This is now line 428 (Section 3.3).

10. Perhaps spend a little time discussing the significance of F. obliquecostata as a strong sea ice indicator. I double checked with your counts, and yes, this does dominate, by a long shot. This shows up in what you plot as the Fragilariopsis group, but the dominance of F. obliquecostata is strong evidence for extensive sea ice. See Crosta et al., 2022, for a summary.[ Crosta, X., Kohfeld, K. E., Bostock, H. C., Chadwick, M., Du Vivier, A., Esper, O., Etourneau, J., Jones, J., Leventer, A., Müller, J., Rhodes, R. H., Allen, C. S., Ghadi, P., Lamping, N., Lange, C., Lawler, K.-A., Lund, D., Marzocchi, A., Meissner, K. J., Menviel, L., Nair, A., Patterson, M., Pike, J., Prebble, J. G., Riesselman, C., Sadatzki, H., Sime, L. C., Shukla, S. K., Thöle, L., Vorrath, M.-E., Xiao, W., Yang, J., 2022, Antarctic sea ice over the past 130,000 years, Part 1: A review of what proxy records tell us, EGUsphere [preprint], https://doi.org/10.5194/egusphere-2022-99.]

Answer: I agree and have highlighted that it dominates the group in the Results, Section 3.1 (line 370). I have also discussed F. obliquecostata as a strong sea ice indicator and presented the reference as suggested (line 369).

Answer 12[th] January: This is now line 467.

11. Figure 3 – I usually prefer greater uniformity in selection of the x-axis scaling. Certainly, a single scale would be inadequate, given the extreme differences in the contribution of different species, but in this figure, every species has its own scale.

Answer: I agree and have made scale amendments in Fig. 3, to have more uniformity where possible.

12. Actinocyclus ingens LAD 0.43-0.5 Ma [Cody, R.D., Levy, R.H., Harwood, D.M. and Sadler, P.M. Thinking outside the zone: High-resolution quantitative diatom biochronology for the

Antarctic Neogene. Palaeogeogr. Palaeoclimatol. Palaeoecol. 260, 92–121 (2008)]. Plus, I would classify it as fairly robust (line 451).

Answer: This reference (line 381; line 545) and description is now added (line 549).

Answer 12th January: The reference is now at line 401, while the description is now at line 643.

13. Lines 470-471 – Thalassiothrix is found in sediment cores from the nearby Sabrina Slope (Holder et al., 2020); I suggest deleting reference to Leventer 1992 – true, about surface sediments, but the Sabrina Slope data are from not so far away. In discussing the habitat for Thalassiothrix, perhaps consider referencing: [P.G. Quilty, K.R. Kerry, and H.J. Marchant, A seasonally recurrent patch of Antarctic planktonic diatoms, Search, pp.48-51, 1985.]

Answer: Holder et al. (2020) do not mention *Thalassiothrix* but *Eucampia antarctica* as a proxy for CDW. I added Quilty et al. 1985 as suggested (line 564).

Answer 12th January: The added reference is now at line 665.

14. Lines 574-575: Perhaps back off this statement; the data are not strong, given the resolution.

Answer: Ok (line 672).

Answer 12th January: This is now erased at line 675. However, the context of the paragraph is changes, and does not consider the PC 4, as before, but down core distribution of Thalassiothrix group.

15. Lines 587-588: What does the sedimentology / x-radiographs suggest? Any evidence for downslope transport?

Answer: The 340-320 cm interval comprises an increase in m silt to clay fraction. And the X-radiographs show laminae. However, these don't necessarily indicate turbidity currents (Rebesco, M, Hernández-Molina, FJ, Van Rooij, D & Wåhlin, A 2014, 'Contourites and associated sediments controlled by deep-water circulation processes: state-of-the-art and future considerations', *Marine geology*), although these are common sediments on the Antarctic margin (Escutia et al. 2003). There could however additionally be a possibility this interval is a turbidite- due to pyrite present (Presti et al. 2011) - which is mentioned in the section (line 687).

Answer January 12th: This is now line 815.

Technical Corrections:

Line 23: Eucampia antarctica terminal/intercalary ratio of what? Low? High? Be specific. 'high' added (line 23).

Now line 31: "..by the increase in the *Eucampia* index (Terminal/Intercalary valve ratio)– an additional proxy for sea ice, which coincides with increases in PC 2."

Line 41: Pritchard. Corrected (line 50; 56).

Now line 51; 58.

Line 50: lowering temperatures – how does this impact productivity? I think you mean that it influences the species composition, but the way it's written implies it influences whether productivity is high or low? This is now corrected by deleting 'lowering temperatures' (line 51).

Now line 59.

Line 55: decrease in production of AABW – a cause or effect? This has been corrected to be both, ice sheet melt causes AABW decrease, but this in turn can affect ice sheet melt (Silvano et al 2018). Line 56;57.

Now line 64.

Line 584: diatom abundance interval instead of diatom interval. This sentence has been erased as per Ref. 2. comment 3. (Line 682).

Now line 809 (erased).

Line 589: pyrite is found instead of pyrites are found. Corrected (line 688).

Now line 821.

Diatom data table, several mis-spellings (Supplement Table S1)

Actinocyclus actinochilus. Corrected.

Coscinodiscus oculoides. Corrected.

Fragilariopsis angulata is now F. rhombica. Corrected.

Fragilariopsis Barbieri. Corrected.

Fragilariopsis pseudonana. Corrected.

Rhizosolenia polydactyla. Corrected.

Rhizosolenia inermis is now Proboscia inermis. Corrected.

Thalassiothrix antarctica. This was written as suggested.

Lea: The location of these changes in the newest version (11th/12th January 2023) are now written beneath each answer to Referee 2, in green.

Referee 2

This article is an interesting and valuable contribution to our understanding of seasonal sea-ice zone dynamics across a full glacial-interglacial cycle. The palaeoenvironmental conditions are reconstructed from a marine sediment core located further south than previous reconstructions of a full glacial-interglacial cycle and thus represents a valuable new data point. The authors use a combination of sedimentological and diatom species assemblage analyses, alongside statistical analysis, to reconstruct the palaeoenvironment of the continental slope region off Adelie Land. This multi-proxy data set is used to investigate the variations in environmental conditions between glacial and interglacial periods, as well as during the glaciation and deglaciation transitions, back to MIS 6.

Overall, the authors do a good job presenting and interpreting the diatom data and show a good appreciation of the limitations and challenges. Particularly those associated with transport and dissolution of diatoms and establishing robust chronologies for Southern Ocean marine sediment cores. Whilst I think this manuscript should be published, there are some areas of concern that I would like to see addressed, and think would help strengthen the manuscripts conclusions.

Thank you for your comments, and your work.

Specific Comments

1.  How did the authors determine which age model details were presented in the main manuscript and which were only in supplemental? For example, in section 2.5 biogenic silica, Si/Al, and IRD are listed as some of the primary data used in age model construction but only the first two have detailed methodologies in the main manuscript. I appreciate that the authors probably don't want to spend too much of the manuscript detailing all of the sedimentology, but I think the current separation could benefit from reassessment.

    Answer: Yes, I agree with this point and have shifted the methodology and results for IRD data in the main manuscript (line 155 and 201, respectively). I have added IRD data into the age model figure (Fig. 2) to species distribution (Fig 3.) and to the final Fig. 5, instead of cell counts.

    Answer 12th January 2023: IRD is now in Methods (line 176) and Results line 358.

2.  Robust age models for Southern Ocean marine sediment cores located so far south are often challenging and I largely agree with the logic used by the authors for the chronology in core Tan_44. However, I think the age model would benefit from additional biomarker evidence (e.g., the last occurrence of *Rouxia leventerae* at the MIS 6-5e boundary). The authors themselves mention the problems with Antarctic ice sheet advance removing the deposited sediments, and the addition of biomarkers would help establish that the interglacial identified as MIS 5e isn't actually an older interglacial.

    Answer: *Rouxia leventerae* wasn't identified in any of the slides. All slides (5-350 cm) were thoroughly analysed. However, I have added this fact in the Age Model section (line 222).

The solution to the question of how we can really know that MIS 5e isn't older, is to undertake additional diatom analysis of the deeper section of core Tan_44, that is, from 350-630 cm, which includes older MIS 6 and MIS 7 interglacial.

Answer 12th January: This is now highlighted in line 370.

3. I have a couple of points on the diatom preparation and counts. Firstly, the authors mention that for species that are highly fragmentary, only the ends were counted, was the same process applied to other pennates? Or were they only counted if >50% of the valve was present? If the latter, how did the authors ascertain they had >50% of the valve for broken valves of species such as *Fragilariopsis cylindrus,* which are linear and isopolar? Secondly, the counts are detailed as >400 valves but it is unclear when the count was stopped, did the entire slide need to be counted, or did the count just continue until the 400 point had been passed? Without details on this it is hard to know how to interpret the diatoms per slide values given in Figure 2. Either way, I would still advise removing this metric from figure 2 and the discussion as it is highly qualitative given the method of slide preparation. Thirdly, I am somewhat confused by the criteria used to include or exclude species/groups from the analyses. Lines 202-3 imply that only species with >2% abundance throughout the core are included in the analysis, but figure 3 and the discussion clearly include species for which this isn't the case (e.g. *Actinocyclus ingens*)? For groups, seemingly the dominant species only needs to have >2% abundance in a single sample, which seems rather inconsistent. I would also caution the authors against grouping by morphology, for example within the *Thalassionema* genus there are substantial differences in environmental preference despite very similar morphologies.

Answer:

3.1 The ends of diatoms were counted only in case of *Thalassiothrix* group, this is written (line 246). For all other diatoms, including pennates, the valves were counted only if they were >50% of the whole (line 245). This means that in the case of Fragilariopsis species, the valve needed to be over >50% the length. Due to the small curve on the outer edge of the Fragilariopsis valve, this wasn't a problem to determine. The isopolar *Fragilariopsis cylindrus* was really rare but also the size of the valve can give us some clue about whether it is >50% of valve.

Answer 12th January: Now line 266.

3.2 I agree the counts are relative, I have counted >400 per slide (line 245). Some slides I counted all of the slide while others I had stopped at a certain point well above 400. I have removed the number of valves per slide from Fig. 2, Figure 3 and from discussion (line 679-682), however I have left the discussion on barren intervals (line 678) and intervals where pyrite is found (line 685-690). I think both are important to mention due to the content/ that is, no content.

Answer 12th January: Results on barren slides now at line 403, and discussion on them now at 811. All discussion on counts is removed- now at line 811. Also in Methods, line 276, and in Results, line 485.

3.3 *Actinocyclus ingens* was found 11% and 3% abundance, within two different samples.

3.4 Species *Thalassiothrix antarctica*, *Thalassiothrix longissima* and *Trixothoxon reinboldii* were grouped together due to very similar morphologies. They all constitute open ocean species with some difference in preference. I present this in Table S1 – for each of these species. However, in the results I present the group is dominated by *Thalassiothrix antarctica* which in number is probably highly underrepresented due to the inability to count its broken very elongated valves. I say this because this species occurs in very high numbers in the sample, in relation to others seen in the core- at 40 and at 270 cm (line 349-353).

Answer 12th January: This is now presented in Methods, line 279, 295 and 298, in Results line 448. The environmental preferences are covered in discussion, line 656.

4. For section 3.4 the authors argument would be strengthened by the inclusion of some p values to show the statistical significance of the regressions. Especially as, to me at least, the $r^2$ values seem rather low for all of the regressions.

Answer: This analysis was changed from regression to correlation, which is more appropriate in this case as we are interested in the strength and direction of the relationship between variables, not the predictive ability of the specific relationship. The p values were added to indicate the significance of the correlation (line .468).

Answer 12th January: This is now line 556.

The paragraph in lines 408-24 feels rather contradictory. The authors seem to suggest both that there is significant reworking of the diatom assemblage, and that the assemblage is a faithful reconstruction of the overlying environmental conditions. The justification for why the authors consider this assemblage to be truly autochthonous needs to be made clearer. Otherwise the reader is left questioning whether the PC1 assemblage can really be trusted any more than the PC3 for reconstructing environmental conditions.

Answer: All assemblages especially on the continental slope and shelf, are reworked to some extent. However, commonly the completely reworked assemblages contain only robust valves of certain species, and these have been defined in sediment, by Taylor and McMinn (1997), and Truesdale and Kellogg (1979). Assemblages which contain other species are therefore considered to contain in situ sedimentation as well as to some extent reworking.

Answer 12th January: this paragraph has only slightly changed from the previous version, by adding an extra species (underlined) as part of the assemblage and further supports the argument.

"The composition of the PC 1 assemblage further suggests that selective species preservation, due to reworking by bottom currents and/or dissolution processes, had been active. The presence of a combination of robust species, e.g., *Eucampia antarctica*, and *Actinocyclus actinochilus,* suggests that some level of reworking of sediments influenced the assemblage composition (Shemesh, Burckle, and Froelich, 1989; Taylor and McMinn 1997). These species have been found within assemblages considered to have been influenced by reworking off Cape Darnley in Prydz Bay (Taylor and McMinn 1997) and the continental slope of the Ross Sea (Truesdale and Kellogg 1979). Reworking is corroborated by the knowledge that the site is currently influenced by the down slope flow of Adélie AABW and along slope currents, including the ASF (Fig. 1; Williams et al. 2008). Furthermore, the presence of unusual abundances of *Thalassiosira lentiginosa* (Fig. 3), a species usually associated with open ocean assemblages (Taylor and McMinn 1997; Truesdale and Kellogg 1979; Crosta et al. 2005) has been associated with dissolution (Shemesh, Burckle, and Froelich,

1989) suggesting that there is some level of dissolution affecting the PC 1 assemblage composition. Such high abundances of *T. lentiginosa* are not observed in modern sediments in the Adélie region (Leventer 1992), or elsewhere within the sea ice zone on the Antarctic margin (Zielinski and Gersonde 1997; Armand et al. 2005; Crosta et al. 2005). Despite the influence of reworking and dissolution, the PC 1 assemblage is still considered to be primarily autochthonous and dominated by *in-situ* deposition associated with open ocean, warmer water, and sea ice edge species. The presence of *Azpeitia tabularis,* and *Asteromphalus hyalinus*, species not commonly associated with reworking or dissolution, further confirms this position."

Technical Corrections

Line 23 - It isn't specified whether it is a high or low *Eucampia* terminal/intercalary ratio associated with PC2. Corrected to 'high *Eucampia antarctica* index' (line 23).

Now "increase in *Eucampia i*ndex" (line 31).

Line 29 - Should be *oliveriana* not *oliverana* (mispelt throughout manuscript). Corrected in text (line 29; line 335; line 434; line 544, 546), Fig 3. And Fig. S5.

Now line 434 (note this species isn't in the main two groups which are now described in abstract); line 525; line 638 and 640.

Line 130-1 - Are the anomlaous spikes identified by statistical comparison to surrounding data or just by eye? 'By eye' is added to text (section 2.3; line 145).

Now line 169.

Line 150 - Core site Tan_68 is shown in Figure 1 but not reference at all in the manuscript. Tan_68 removed from Fig. 1.

Line 157-8 - The lines showing the average position of the monthly sea-ice edge are not explained in the figure caption. I assume the lines are sourced from Fetterer et al. (2017) and the blue shading from Spreen, Kaleschke & Heygster (2008) but this also isn't made clear. This has been made clear in Fig 1. caption and in text (line 109-110).

Now line 121.

Line 165 - There is no explanation in the main manuscript on what the D and R in the %microfossil row stand for. Explanation added in caption of Table 1.

Now: the microfossil % is now completely erased. (Line 361), as explained in Referee 1 Answer 6.

Line 169 - Only two radiocarbon dates are mentioned but Figure 2 and Table S2 both contain 4. This has been explained in text (Age Model; line 214-217).

Now line 372 explains why these are left out of interpretation.

Line 374 - The PC3 and biogenic silica regression has an $r^2$ >0.1. This section is changed – see answer to question 4, above.

Line 451 - I would consider *A. ingens* to also be fairly robust so don't think the except is necessary. This has been corrected 'except' is replaced by 'including' (line 549).

Now line 643.

Line 589 - Should be "pyrite is". Corrected (line 688).

Now line 821.

Line 607 - kyrs as one word. Corrected in text (line 705), and figures Fig. 2, Fig. 3 and Fig. 5.

Now line 840.

---

## Author Response (AR2)

**Public justification (visible to the public if the article is accepted and published)**:
Dear Dr. Pesjak,

I believe that you answered adequately most of the reviewers' comments as well as my editorial comments. The manuscript improved strongly. I however still have some concerns on the main text and on the supplementary that must be taken into consideration before publication. Most of these comments relate to taxonomic issues and figures.

Thank you.

Main text
Lines 67-70 : For taphonomic processes, I would refer to Warnock and Scherer, 2015.

Answer: I have added Warnock and Scherer (2015) as suggested, and furthermore, it was added to the Discussion, line 491:
"The presence of a combination of robust species, e.g., *Eucampia antarctica*, and *Actinocyclus actinochilus,* suggests that some level of reworking of sediments (Shemesh, Burckle, and Froelich, 1989; Taylor and McMinn 1997) or dissolution (Warnock and Scherer 2015) influenced the assemblage composition. These species have been found within assemblages considered to have been influenced by reworking offshore Cape Darnley in Prydz Bay (Taylor and McMinn 1997) and the continental slope of the Ross Sea (Truesdale and Kellogg 1979)."

Lines 73-75 : If you want to be a bit more exhaustive and less regional, you can cite: Peck et al., 2015; Mezgec et al., 2017; Torricella et al., 2021 for a few references dealing with coastal Holocene diatom studies.

Answer: Completed as suggested.

Lines 165-168 : Please detail which calibration curve you used : Marine13 (or 20?) or SHCal13 (or 20?)?

Answer: I used Marine13. This is now added, line 166:
"..The raw radiocarbon ages were calibrated using CALIB, version 7.1, Marine13 calibration curve (Reimer et al. 2013) and the regional variation to the global marine reservoir correction, ΔR, of 830 yrs ±200 yrs,.."

Lines 175-179 : Typo errors. Please change to « ....The age model of Tan_44 is based on the facies model and two radiocarbon dates from the top 25 cm of the core,

using the premises that variability in facies, including large changes in productivity proxies (biogenic silica, Si/Al and Ba/Ti) and IRD content, present glacial…. »

Answer: Completed.

Lines 198-207 (and elsewhere in the manuscript where applicable) :
- Fragilariopsis rhombica is not a sea ice species per se (Armand 2005 ; Esper 2010) rather a cold water species with a maximum relative abundance in modern sediments along a large sea ice range and presence even at no sea ice. I however reckon its presence in this group may not change much your interpretations.

Answer: As suggested, I have now removed *Fragilariopsis rhombica* from the sea ice Fragilariopsis group, in line suggested and elsewhere in text. It is present in trace amounts, and its removal does not influence relative abundance (%) of Fragilariopsis sea ice group species.

- Your taxonomy for the genus Rhizosolenia is outdated and questionable. Rhizosolenia styliformis is sea ice restricted, thriving on the continental shelf (Ligowski's papers). Rhizosolenia antennata with two processes is called R. antennata var antennata (and not Rhizosolenia twin process antennata). Your Rhizosolenia antennata is probably R. antennata var semispina (with one process and pointed otaria extending onto the process; Sundstrom, 1986; Priddle 1990; Armand 2001). I suspect you have confused R. styliformis with R. antennata var semispina. Please note that Rhizosolenia hebetata is not present in the Southern Ocean as it is restricted to the northern hemisphere (Hasle and Syversten, 1997). This is probably R. simplex or R. species A (Armand 2001, etc…). Rhizosolenia setigera is absent from polar waters (Hasle and Syversten, 1997). This is probably R. simplex or R. species A. Your Rhizosolenia polydactyla is probably R. polydactyla var polydactyla (rounded otaria extending onto the prcess). Proboscia intermis is P. inermis (wrong spelling).

Answer:
I have now corrected all, as suggested (line 202, 215, 334, 583, and 538, Fig. 3 caption, Table S1 and Table S5). Note, Armand (2001) cited online is Armand and Zielinski 2011 as written on paper itself. I cite this paper now in text, line 583 and in caption of Table S1.
-Rhizosolenia (twin process) antennata is now changed to Rhizosolenia antennata var. antennata.
-Rhizosolenia antennata is now changed to Rhizosolenia antennata var. semispina.
-Rhizosolenia styliformis is now changed to Rhizosolenia antennata var. semispina (adding the old Rhizosolenia antennata does not change the overall percentage of former R. styliformis).

-Rhizosolenia hebetata is now changed to R. simplex.
-Rhizosolenia setigera is now changed to R. simplex.
(The joining of R. hebetata and R. setigera into R. simplex doesn't change their overall abundance, these are very rare species).
-Rhizosolenia polydactyla is now changed to R. polydactyla var polydactyla.
-The correct spelling of Proboscia inermis is added (line 204).

Lines 252 (and throughout the manuscript) : Please spell up the genus name when starting a sentence.

Answer: I haven't found this mistake at line 252. I have checked throughout the manuscript – and corrected at line 579 (Fragilariopsis kerguelensis).

Line 564 : Recent publication by Jones et al. 2022 in CP.

Answer: Reference is now added, line 620:
"This is consistent with gradual cooling reaching a maximum at the end of MIS 2, as seen in Antarctic ice cores (Jouzel et al. 1993) and Sea Surface Temperatures from global sediment cores (Kohfeld & Chase 2017), including records based on diatom assemblages from the Southern Ocean north of 56 °S (Crosta et al. 2004; Chadwick et al. 2022; Jones et al. 2022)."

Figures 2 and 3 : Facies 2A from 50cm to 25cm covers the 18-14 ka BP period on figure 3, but present ages between 22-16 ka BP in figure 2. And goes probably beyond 22 ka BP as the deeper date is 2/3 down this facies. Paradoxical.

Answer: Figures 2 and 3 are now aligned with respect to the age at 25 cm: it has been aligned with the 16 ka, C-14 date at 25 cm. The older line in Fig. 3 has been removed, consistent with Fig. 2.

Figure 3 : Please explain what are the two blue colors for the Eucampia index in the figure caption.

Answer: Added to caption to Fig. 3 as suggested, line 380:
".. Eucampia index was also not included in statistical analysis, its distribution in dark blue shows where total Eucampia antarctica counts are >100 valves per sample, while the light blue areas show samples with <100 counts".

Figure 5 : Relocate the Eucampia index X axis down the figure to lighten the text and scales presented in the upper part of the figure. Make sure that all labels are of the same police and font size. Some axes' labels are not even aligned.

Answer: I relocated Eucampia index X axis and made all the labels the same size (but left the sublabels i.e. more sea ice, smaller). I aligned other axes' labels better.

Figure 6 : Try to make the text a bit more readable as there is more text now in each box. Please use color.

Answer: I reduced the amount of text, and produced more legend to tell the story through illustrations (i.e., biodiversity, high productivity, increase in CDW). I also coloured the illustration, as suggested. However, I would prefer the black and white sketch, over the coloured version. I have presented both versions in the tracked version of manuscript but left only the black and white in the clean version.

Supplement material

Table S1 : Please define « op » in the footnote of the table (as done for W, etc…). Fragilariopsis angulata is now F. rhombica. Change the Rhizosolenia species' names to accomodate my comment above.

Answer:
-op has now been defined.
-Fragilariopsis angulata has been erased from the table, as it no longer is within the Fragilariopsis sea ice group.
-Rhizosolenia species have been renamed as per suggested above, and an additional reference added.

Table S4 : Coscinodiscus vulnificus was renamed Thalassiosira vulnifica a long time ago (Harwood, 1992). Please use Thalassiosira oliverana or Thalassiosira oliveriana throughout the tables and main text, but not both. Change the Rhizosolenia species' names to accomodate my comment above.

Answer:
-Coscinodiscus vulnificus is now changed to Thalassiosira vulnifica (and shifted in alphabetical order);
-T. oliveriana was already written in S4, but I corrected this in Table S2 and in Table 3. I have now increased the font size in Table S4.
-Rhizosolenia species have been changed as per all suggested and shifted in alphabetical order.